# Uncertainty-Aware Multi-Objective Reinforcement Learning-Guided Diffusion Models for 3D De Novo Molecular Design

**Lianghong Chen[1], Dongkyu Eugene Kim[2], Mike Domaratzki[1], Pingzhao Hu[1,2]\***
[1]Department of Computer Science, Western University, London, ON, Canada
[2]Department of Biochemistry, Western University, London, ON, Canada
`lchen776@uwo.ca, phu49@uwo.ca`

## Abstract

Designing de novo 3D molecules with desirable properties remains a fundamental challenge in drug discovery and molecular engineering. While diffusion models have demonstrated remarkable capabilities in generating high-quality 3D molecular structures, they often struggle to effectively control complex multi-objective constraints critical for real-world applications. In this study, we propose an uncertainty-aware Reinforcement Learning (RL) framework to guide the optimization of 3D molecular diffusion models toward multiple property objectives while enhancing the overall quality of the generated molecules. Our method leverages surrogate models with predictive uncertainty estimation to dynamically shape reward functions, facilitating balance across multiple optimization objectives. We comprehensively evaluate our framework across three benchmark datasets and multiple diffusion model architectures, consistently outperforming baselines for molecular quality and property optimization. Additionally, Molecular Dynamics (MD) simulations and ADMET profiling of top generated candidates indicate promising drug-like behavior and binding stability, comparable to known Epidermal Growth Factor Receptor (EGFR) inhibitors. Our results demonstrate the strong potential of RL-guided generative diffusion models for advancing automated molecular design. The implementation is available at `https://github.com/Kyle4490/RL-Diffusion`.

## 1 Introduction

The design of novel molecules with desirable properties is fundamental to drug discovery, materials science, and molecular engineering [1, 2]. Deep generative models have emerged as promising tools for automated molecular design, enabling efficient exploration of vast chemical spaces that are difficult to navigate manually [3–5]. Among these, diffusion models have demonstrated remarkable capabilities in capturing complex molecular distributions and generating diverse, high-quality samples [6]. However, most existing studies have focused on satisfying basic chemical validity constraints without explicitly controlling multiple drug-relevant property objectives [7]. Achieving reliable multi-objective optimization remains a major challenge, particularly for therapeutically important targets such as the Epidermal Growth Factor Receptor (EGFR), a key receptor protein involved in cancer progression and drug resistance [8]. The ability to efficiently design molecules with optimal combinations of drug-likeness, synthetic accessibility, and binding affinity to receptor proteins is critical for advancing precision drug development.

---

\*Corresponding author. Department of Computer Science, Western University, 1400 Western Road, London, Ontario N6G 2V4, Canada. Email: phu49@uwo.ca (P.H.)

39th Conference on Neural Information Processing Systems (NeurIPS 2025).

To address the challenges of property optimization, prior research has explored strategies such as flow matching [9, 10] and energy-guided generation [11, 12]. Although these methods provide effective solutions under specific conditions, they generally require explicit and differentiable reward functions, making them poorly suited for handling black-box objectives, such as Quantitative Estimate of Drug-likeness (QED) [13], Synthetic Accessibility Score (SAS) [14], and binding affinity [15], which are predicted by external computational tools and are critical for the success of drug development. [7]. In contrast, Reinforcement Learning (RL) has emerged as a particularly versatile alternative due to its strong flexibility in managing non-differentiable rewards and dynamically balancing multiple property objectives [16]. RL has been widely applied to guide the optimization of molecular generation models such as Recurrent Neural Networks (RNNs) [17], Variational Autoencoders (VAEs) [18–20], and Transformers [21]. However, these advancements have largely focused on 1D string representations such as Simplified Molecular Input Line Entry System (SMILES) [22] and 2D molecular graphs. RL-guided optimization of diffusion models to directly generate de novo 3D molecules has not yet been fully explored. Generating de novo 3D molecules with precise molecular geometries and chemical interactions is essential for drug development-relevant downstream tasks like molecular docking and Molecular Dynamics (MD) [23], which cannot be achieved using 1D or 2D molecular representations.

In this study, we propose an uncertainty-aware multi-objective RL framework to guide the optimization of 3D molecular generative diffusion models. Our approach leverages property uncertainties predicted by surrogate models to guide reward assignment and stabilize optimization under complex multi-objective constraints. We evaluate our method on three widely used molecular datasets: the QM9 Quantum Chemistry Dataset, the ZINC15 Molecular Library, and the PubChem Compound Database, comparing it against multiple strong baselines and conducting comprehensive ablation studies. To further validate the scalability of our method, we also apply it to different diffusion model architectures. Moreover, in addition to considering key molecular properties such as binding affinity, we further assess the practical drug development potential of the generated candidate molecules through MD simulations and Absorption, Distribution, Metabolism, Excretion, and Toxicity (ADMET) property assessments [24], benchmarking against known EGFR inhibitors. Our results highlight the strong potential of RL-guided diffusion models to design candidate molecules with superior stability and drug-likeness, providing a new paradigm for diffusion model-driven de novo 3D molecular design.

Our contributions are summarized as follows:

- We propose the first end-to-end framework that integrates RL, diffusion models, and uncertainty quantification for 3D molecular generation with multi-objective optimization.

- We design a novel reward function that integrates uncertainty quantification with three auxiliary components, including a reward boosting mechanism, a diversity penalty, and a dynamic cutoff strategy. This design balances optimization trade-offs among multiple objectives and addresses challenges such as reward sparsity and mode collapse when applying RL to optimize diffusion models. As a result, both the overall quality of generated molecules and the number of candidates that satisfy all property requirements are improved.

- We conduct extensive experiments on three benchmark datasets, comparing against multiple State-Of-The-Art (SOTA) baselines and performing comprehensive ablation studies to assess the contribution of each component. To validate the generality of our framework, we further apply it to different 3D molecular diffusion model architectures.

- We demonstrate the real-world potential of our approach through MD simulations and ADMET profiling of generated candidate molecules, benchmarking against known EGFR inhibitors.

## 2 Related Works

**De novo 3D molecular generation.** Early approaches to 3D molecular generation, such as G-SchNet [25] and E(3)-Normalizing Flows (E-NF) [26], applied autoregressive generation and equivariant flows to model spatial symmetries. More recently, diffusion models have emerged as a promising paradigm. Equivariant Diffusion Model (EDM) [27] introduced E(3)-equivariant denoising to better capture molecular geometries. Geometric Latent Diffusion Model (GeoLDM) [28] extended this by applying latent space diffusion. Molecular Diffusion Model (MDM) [29] and Generative Force

Matching Diffusion Model (GFMDiff) [30] further incorporated physics-based constraints to improve structural realism and diversity.

**RL-guided generative model optimization.** Prior works on RL-guided diffusion models have mainly focused on image generation tasks with single-objective optimization. Fan et al. proposed Shortcut Fine-Tuning with Policy Gradient (SFT-PG) [31] to reduce distributional mismatch and improve sample quality. Denoising Diffusion Policy Optimization (DDPO) [32] treated denoising as a multi-step decision process and introduced DDPO-Importance Sampling (IS) and DDPO-Score Fine-tuning (SF) to incorporate preference-conditioned rewards. DPOK [33] further added Kullback-Leibler (KL) [34] regularization to stabilize training and enhance alignment.

**Multi-objective optimization methods.** Existing multi-objective optimization methods can broadly be grouped into scalarization-based, constraint-based, gradient-based, and uncertainty-based approaches. Scalarization methods, such as Weighted Sum (WS) [35], Product Of Objectives (POO) [36], Max-Min Method (MMM) [37], and Linear Scalarization with Dynamic Weights (LSDW) [38], convert the problem into single-objective optimization but require careful weight tuning. Constraint-based methods, including Normalized Manhattan Distance (NMD) [39], NMD-WS [40], Compromise Programming (CP) [41], and Penalty Function Methods (PFM) [42], enforce objectives as constraints but can be unstable when conflicts arise. Gradient-based methods like Singular Value Decomposition (SVD) [43], PCGrad [44], CAGrad [45], and GradVac [46] adjust gradients to resolve conflicts locally, but cannot model the global Pareto front. Recently, uncertainty-based methods have emerged. Chen et al. [47] proposed an uncertainty-quantified genetic algorithm for molecular design. Other methods include Upper Confidence Bound (UCB) [48], Expected Improvement (EI) [49], Maximum Variance Criterion (MVC) [50], and Bayesian Optimization by density-Ratio Estimation (BORE) [51], which have shown promise but remain underexplored in RL-guided generative modeling.

## 3   Methodology

This section presents the core components of our framework, including the diffusion model backbone, surrogate models, and RL-guided optimization. The overall architecture of our framework is illustrated in Fig. 1. Detailed explanations, mathematical formulations, and derivations are deferred to Appendix A.

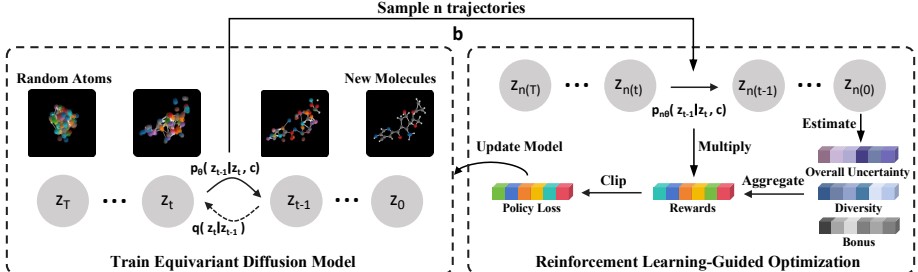

Figure 1: **Architecture diagram of the RL-guided diffusion model.** (a) Conditional EDM is trained to generate 3D molecules from random atoms, conditioned on target properties. The model learns a forward noising process $q(z_t|z_{t-1})$ and a reverse denoising process $p_\theta(z_{t-1}|z_t, c)$. (b) In the RL phase, the pre-trained conditional EDM is used to generate n molecules by sampling full denoising trajectories. The transition probabilities $p_\theta$ at each timestep are recorded. For each generated 3D molecule, structural diversity is computed, and a reward boost is applied if the molecule satisfies validity, uniqueness, and novelty criteria. In parallel, surrogate models are used to estimate the probability that each molecular property exceeds its cutoff. These probabilities are multiplied to compute the overall success probability (termed "overall uncertainty", i.e., the likelihood that the molecule satisfies all desired property thresholds.) Rewards are constructed by combining the overall uncertainty, the bonus, and the diversity. These rewards are multiplied along trajectories and used, together with transition log-probabilities, to compute a clipped policy gradient loss for updating the model.

### 3.1 Conditional EDM Backbone

Since most current 3D molecular diffusion models build upon EDM, we select it as our backbone to ensure compatibility with future architectural extensions. EDM consists of two core components:

**Forward diffusion.** The forward process adds Gaussian noise to the atomic coordinates and features $\mathbf{x} = (\mathbf{r}, \mathbf{h})$, where $\mathbf{r} \in \mathbb{R}^{M \times 3}$ denotes atomic positions and $\mathbf{h} \in \mathbb{R}^{M \times d}$ denotes atom features. At each timestep $t \in \{1, \ldots, T\}$, noisy latents are sampled via $q(\mathbf{z}_t \mid \mathbf{x}) = \mathcal{N}(\mathbf{z}_t; \alpha_t \mathbf{x}, \sigma_t^2 \mathbf{I})$, where $\alpha_t$ and $\sigma_t$ are schedule parameters. This defines a Markov chain of corrupted states $\{\mathbf{z}_t\}$ used to train the denoising model and establish a reverse sampling chain.

**Backward diffusion.** The reverse process approximates the denoising distribution $p(\mathbf{z}_{t-1} \mid \mathbf{z}_t, c)$ using a parameterized Gaussian:

$$p(\mathbf{z}_{t-1} \mid \mathbf{z}_t, c) = \mathcal{N}\left(\mathbf{z}_{t-1}; \mu_\theta(\mathbf{z}_t, t, c), \sigma_t^2 \mathbf{I}\right), \tag{1}$$

where $\mu_\theta(\mathbf{z}_t, t, c) = \frac{1}{\alpha_t} \mathbf{z}_t - \frac{\sigma_t}{\alpha_t} \hat{\boldsymbol{\epsilon}}_\theta(\mathbf{z}_t, t, c)$, $c$ is the condition vector concatenated to the atom features, and $\hat{\boldsymbol{\epsilon}}_\theta$ is the predicted noise from an E(n)-equivariant Graph Neural Network (EGNN) [52].

### 3.2 Surrogate Models for Multi-objective Uncertainty Quantification

The surrogate models quantify the uncertainty of molecular property satisfaction during RL-guided optimization. Each model estimates the likelihood that a molecule exceeds a predefined threshold for a specific property, providing a smooth reward signal for multi-objective policy learning.

**Single-property uncertainty modeling.** We employ Chemprop's Directed Message Passing Neural Network (D-MPNN) [53] as the surrogate predictor. Each surrogate model is trained to estimate the predictive mean $\mu(m)$ and variance $\sigma^2(m)$ of a molecular property for a given molecule $m$. Assuming the prediction follows a Gaussian distribution, we define the uncertainty-aware reward for a single property as:

$$U_{\text{single}}(m; \delta) = \eta \int_\delta^\infty \frac{1}{\sigma(m)\sqrt{2\pi}} \exp\left(-\frac{1}{2}\left(\frac{x - \mu(m)}{\sigma(m)}\right)^2\right) dx, \tag{2}$$

where $\delta$ is the target threshold, and $\eta \in \{+1, -1\}$ indicates task directions, with $\eta = +1$ for properties where higher values are preferred (e.g., QED) and $\eta = -1$ for properties where lower values are preferred (e.g., SAS and binding affinity).

**Multi-objective reward aggregation.** Since each property is modeled independently and assumed to be conditionally independent given the molecule, the overall multi-objective reward is defined as the joint satisfaction probability:

$$U_{\text{multi}}(m; \delta_1, \ldots, \delta_k) = \prod_{i=1}^k U_{\text{single}}^i(m; \delta_i), \tag{3}$$

where $k$ is the number of target properties and $U_{\text{single}}^i$ is the reward for property $i$. The aggregated score $U_{\text{multi}}(m; \cdot) \in [0, 1]$ provides a smooth and interpretable signal that favors molecules likely to satisfy all property objectives simultaneously.

### 3.3 RL-guided Optimization

RL-guided optimization process consists of three key components: trajectory sampling, reward design, and policy update.

**Trajectory sampling.** At each episode, we sample $n$ molecules and record their generation trajectories $\{\mathbf{z}_T, \mathbf{z}_{T-1}, \ldots, \mathbf{z}_0\}$. In RL, an explicit probability density is required for gradient estimation. Therefore, the reverse transition in Equation (1) is rewritten in its Probability Density Function (PDF) [54] form as:

$$p_\theta(\mathbf{z}_{t-1} \mid \mathbf{z}_t, c) = \frac{1}{(2\pi\sigma_t^2)^{d/2}} \exp\left(-\frac{1}{2\sigma_t^2} \|\mathbf{z}_{t-1} - \mu_\theta(\mathbf{z}_t, t, c)\|^2\right), \tag{4}$$

where $\sigma_t^2$ is the variance at timestep $t$, and $d$ is the dimensionality of the latent representation. This formulation defines the likelihood of each denoising step in the Markov chain and serves as the basis for RL policy gradient estimation.

**Reward design.** We design an uncertainty-driven reward function. Considering Equation (3), the total reward for each generated molecule $m$ is defined as:

$$R_{\text{total}}(m; \delta_1, \ldots, \delta_k, t_{\text{episode}}) = U_{\text{multi}}(m; \delta_1, \ldots, \delta_k) \cdot R_{\text{bonus}}(m) - \lambda(t_{\text{episode}}) \cdot D(m), \quad (5)$$

where $t_{\text{episode}}$ denotes the RL training episode index, $R_{\text{bonus}}(m)$ is determined by the validity, uniqueness, and novelty of $m$ (the more conditions satisfied, the higher bonus obtained), $\delta_i$ $(1 \leq i \leq k)$ is a property threshold dynamically updated based on a moving average of the corresponding property values from previously generated molecules, and $D(m)$ denotes the average Tanimoto similarity between $m$ and the other molecules within the same batch. To encourage early-stage exploration and later-stage exploitation, the penalty weight decays over training episodes as: $\lambda(t_{\text{episode}}) = \lambda_0 e^{-\alpha t_{\text{episode}}}$.

**Policy update.** We adopt a Proximal Policy Optimization (PPO)-style [55] strategy to update the parameters of the diffusion model. The objective is to minimize the clipped surrogate loss:

$$\mathcal{L}_{\text{PPO}} = -\mathbb{E}_{m \sim p_\theta} \left[ \min \left( r(m) \cdot R_{\text{total}}(m), \, \text{clip}(r(m), \, 1 - \epsilon, \, 1 + \epsilon) \cdot R_{\text{total}}(m) \right) \right], \quad (6)$$

where $r(m) = \frac{p_\theta(m)}{p_{\theta_{\text{old}}}(m)}$ denotes the likelihood ratio between the current policy and the previous policy and $\epsilon$ denotes the clipping range.

# 4 Experiments

## 4.1 Experimental Settings

This section describes the experimental design of our study, including datasets and data processing, surrogate model training, diffusion model training, RL-guided diffusion model training, baseline comparison, ablation study design, MD simulations and ADMET property prediction, evaluation, and computational resources. Detailed settings, dataset descriptions, hyperparameter configurations, and implementation details can be found in Appendix B.

**Datasets and Data Preprocessing.** We collected 3D molecular structures and SMILES strings of the three datasets. RDKit [56] was used to compute QED and SAS scores, and AutoDock Vina [57] was applied to evaluate binding affinity to EGFR. Each molecule is associated with a SMILES string, 3D structure, QED score, SAS score, and binding affinity. The datasets were split into training (80%), validation (10%), and test (10%) sets. For diffusion model training, molecules were split by species grouping. For surrogate model training, molecules were split by molecular scaffold. All models were trained and evaluated using standard train/validation/test splits.

**Train surrogate models.** We use Chemprop to train surrogate models predicting molecular properties from SMILES. A separate model is trained for each property in each dataset, yielding nine models in total. Hyperparameters are selected via grid search.

**Build diffusion models.** Conditional EDM is trained on 3D molecular structures and property distributions, using the original hyperparameters. GeoLDM and GFMDiff are also trained for comparative analysis.

**RL-guided diffusion model training.** The model is trained by sampling property combinations exceeding predefined cutoffs from the training set as input conditions. It generates molecules conditioned on these properties. Generated molecules are evaluated for validity, uniqueness, novelty, Tanimoto diversity, and uncertainty to compute rewards. Rewards are used to update model parameters via policy gradient.

**Baseline comparison.** We adapt four RL-guided diffusion baselines for image generation (SFT-PG, DDPO-SF, DDPO-IS, DPOK) to the molecular domain. We reformulated the task into a single-objective setting to fit these baselines. All baselines were trained across all datasets. Performance was compared against a vanilla diffusion model and our framework.

**Ablation study design.** Our ablation study includes two parts. First, we replace the multi-objective optimization module with four categories of alternative strategies: scalarization-based (WS, POO, MMM, LSDW), constraint-based (NMD, NMD-WS, CP, PFM), gradient-based (SVD, PCGrad,

CAGrad, GradVac), and uncertainty-based (UCB, EI, MVC, BORE). Second, we evaluate the effects of disabling key components: reward boosting, diversity penalty, and dynamic cutoff adjustment.

**MD simulations and ADMET property prediction.** MD simulations were performed using AmberTools [58] and OpenMM [59]. Ligands and receptors were parameterized with ff14SB [60] and GAFF [61], solvated with TIP3P water [62], and simulated under standard conditions (300 K, 1 bar) for a total physical simulation time of 4,000 ps (approximately 1,000,000 steps). ADMET properties were predicted with ADMET-AI [24].

**Evaluation.** Surrogate models were evaluated using the coefficient of determination ($R^2$, measuring goodness of fit) [63], residual analysis, and Area Under the Calibration Error curve (AUCE, measuring predictive uncertainty calibration) [64]. To evaluate generation performance, 2,000 molecules were generated per run across three independent runs. Evaluation metrics included validity, uniqueness, novelty, molecular stability, atomic stability, and proportion of desired candidates.

**Device.** All experiments were conducted on NVIDIA A100 GPUs with 80 GB memory.

## 4.2 Results and Discussion

This section presents and discusses the experimental results, including surrogate model evaluation, diffusion model performance with and without RL-guided optimization, comparisons with SOTA baselines, ablation studies, and MD and ADMET analyses. Additional results, including detailed surrogate model performance, diffusion model performance across architectures, extended examples of generated molecules, and further analysis and explanation are provided in Appendix C.

### 4.2.1 Surrogate Models

We assess the prediction accuracy and uncertainty calibration of the trained surrogate models. The models achieve strong regression performance across all datasets and properties, with $R^2$ values of 0.95–0.99 for QED and SAS, and 0.86–0.88 for binding affinity. Residual plots and predictive variance analysis confirm that model uncertainty correlates with prediction errors. Uncertainty calibration is evaluated using reliability diagrams. The AUCE remains low across datasets and properties (0.02–0.10), indicating well-calibrated uncertainty estimates. These results demonstrate that surrogate models not only accurately predict molecular properties but also provide reliable uncertainty estimates, validating their use as a guidance signal for RL fine-tuning of diffusion models.

### 4.2.2 RL-guided Diffusion Models

Fig. 2 shows the training process of the RL-guided diffusion models. As shown in Fig. 2a–c, the quality metrics, especially validity and molecular stability, are gradually improved over the course of training. The reward curves (Fig. 2d–f) grow steadily and eventually stabilize, indicating convergence. These results confirm that our RL framework enables effective and stable optimization.

Table 1 compares our uncertainty-aware RL-guided diffusion model with state-of-the-art RL baselines and a vanilla diffusion model across all datasets. On QM9, our method achieves the best overall performance, with the highest validity (98.17%), VUN score (88.90%), and top molecule percentage (28.33%). It improves validity by over 9% compared to RL baselines and demonstrates better balance across validity, uniqueness, and novelty. On ZINC15, our model attains near-perfect validity (99.02%) and outperforms all baselines across metrics, including a 33.40% top molecule ratio. This aligns with the drug-like nature of ZINC15, which matches our property objectives better than the small organic compounds in QM9. On PubChem, despite its structural complexity, our method maintains 100% uniqueness and novelty, and outperforms baselines by a large margin. The relatively lower validity (16.23%) is attributed to the mismatch between pre-training on simpler molecules and the high diversity of PubChem. The discrepancy between training and evaluation (Fig. 2c vs. Table 1) arises from the evaluation using a larger sample size (2000 vs. 128 molecules). Overall, our method consistently outperforms all baselines across datasets, confirming its effectiveness and generalizability.

As shown in Fig. 3, we compared the property distributions of 2,000 valid molecules generated with and without RL guidance across three trials. RL-guided models consistently improved QED and reduced SAS and binding affinity, indicating better drug-likeness, synthetic accessibility, and binding potential. The extent of improvement varied across datasets due to intrinsic property and structure

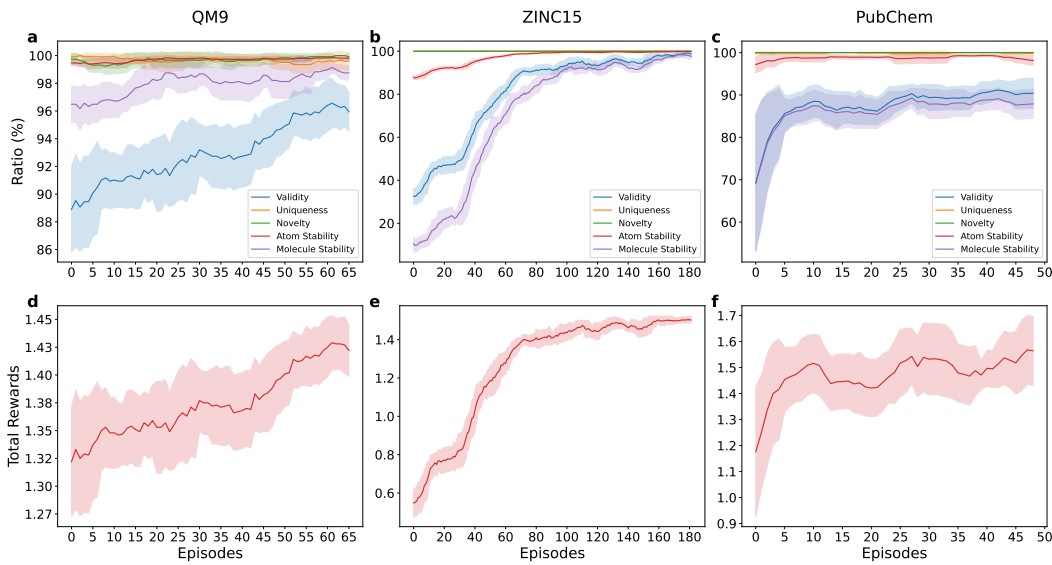

Figure 2: **RL-guided diffusion model training performance across three datasets.** (a–c) Evolution of key molecular quality metrics (validity, uniqueness, novelty, atom stability, and molecular stability) over RL training episodes, where the vanilla diffusion models were pre-trained on QM9, ZINC15, and PubChem, respectively. (d–f) Corresponding total reward curves showing convergence trends.

Table 1: **Performance of baseline models and our method across three molecular datasets.**

| Dataset | Method | Val (%) (↑) | Uni (%) (↑) | Nov (%) (↑) | VUN (%) (↑) | ASta (%) (↑) | MSta (%) (↑) | Top (%) (↑) |
|---|---|---|---|---|---|---|---|---|
| QM9 | W/O RL | 88.55 ± 0.65 | **97.57** ± **0.30** | 99.75 ± 0.15 | 86.19 ± 1.02 | 99.34 ± 0.11 | 95.90 ± 0.34 | 25.17 ± 1.17 |
| | SFT-PG | 88.57 ± 1.33 | 96.80 ± 0.33 | 99.80 ± 0.15 | 85.57 ± 1.60 | 99.24 ± 0.10 | 95.62 ± 0.58 | 25.58 ± 1.65 |
| | DDPO-SF | 88.65 ± 1.05 | 97.39 ± 0.48 | 99.79 ± 0.20 | 86.16 ± 1.59 | 99.25 ± 0.09 | 95.50 ± 0.34 | 25.65 ± 1.51 |
| | DDPO-IS | 88.82 ± 1.03 | 96.59 ± 0.71 | 99.39 ± 0.04 | 85.27 ± 1.30 | 97.31 ± 0.09 | 86.10 ± 0.64 | 25.77 ± 1.29 |
| | DPOK | 88.10 ± 0.37 | 97.52 ± 0.42 | **99.81** ± **0.15** | 85.75 ± 0.80 | 99.18 ± 0.03 | 95.28 ± 0.18 | 25.20 ± 1.44 |
| | **Ours** | **98.17** ± **0.07** | 90.90 ± 0.72 | 99.63 ± 0.04 | **88.90** ± **0.68** | **99.87** ± **0.03** | **99.17** ± **0.27** | **28.33** ± **0.61** |
| ZINC15 | W/O RL | 30.05 ± 1.34 | 100.00 ± 0.00 | 100.00 ± 0.00 | 30.05 ± 1.34 | 88.36 ± 0.40 | 12.00 ± 1.33 | 8.02 ± 0.46 |
| | SFT-PG | 41.25 ± 1.48 | 100.00 ± 0.00 | 100.00 ± 0.00 | 41.25 ± 1.48 | 91.71 ± 0.08 | 25.55 ± 0.65 | 10.43 ± 0.73 |
| | DDPO-SF | 30.25 ± 1.56 | 100.00 ± 0.00 | 100.00 ± 0.00 | 30.25 ± 1.56 | 88.37 ± 0.40 | 11.97 ± 1.41 | 8.05 ± 0.61 |
| | DDPO-IS | 30.47 ± 1.39 | 100.00 ± 0.00 | 100.00 ± 0.00 | 30.47 ± 1.39 | 88.35 ± 0.44 | 12.02 ± 1.57 | 8.13 ± 0.60 |
| | DPOK | 30.13 ± 2.28 | 100.00 ± 0.00 | 100.00 ± 0.00 | 30.13 ± 2.28 | 88.42 ± 0.61 | 12.01 ± 1.38 | 8.02 ± 0.78 |
| | **Ours** | **99.02** ± **0.46** | 99.75 ± 0.06 | 100.00 ± 0.00 | **98.77** ± **0.49** | **99.86** ± **0.03** | **98.08** ± **0.63** | **33.40** ± **0.89** |
| PubChem | W/O RL | 7.18 ± 4.78 | 99.67 ± 0.65 | 100.00 ± 0.00 | 7.17 ± 4.80 | 94.51 ± 0.16 | 38.18 ± 0.92 | 2.23 ± 1.65 |
| | SFT-PG | 7.47 ± 1.40 | 99.57 ± 0.85 | 100.00 ± 0.00 | 7.44 ± 1.37 | 82.99 ± 0.49 | 33.25 ± 0.34 | 2.03 ± 0.76 |
| | DDPO-SF | 7.98 ± 2.96 | 100.00 ± 0.00 | 100.00 ± 0.00 | 7.98 ± 2.96 | 94.49 ± 0.98 | 44.22 ± 0.32 | 2.40 ± 0.37 |
| | DDPO-IS | 10.50 ± 6.19 | 99.90 ± 0.20 | 100.00 ± 0.00 | 10.48 ± 6.16 | 95.36 ± 0.99 | 45.37 ± 1.85 | 2.52 ± 1.22 |
| | DPOK | 7.65 ± 1.75 | 99.67 ± 0.64 | 100.00 ± 0.00 | 7.62 ± 1.76 | 94.51 ± 0.20 | 38.17 ± 0.64 | 2.42 ± 0.46 |
| | **Ours** | **16.23** ± **9.72** | 100.00 ± 0.00 | 100.00 ± 0.00 | **16.23** ± **9.72** | **99.04** ± **0.13** | **88.65** ± **0.59** | **2.97** ± **1.60** |

Note: Each model generates 2,000 molecules per run. Results are averaged over three independent runs and reported as mean ± 95% confidence interval. "W/O RL" denotes vanilla diffusion models without RL. "Val", "Uni", and "Nov" represent the percentages of valid, unique, and novel molecules, respectively. "VUN" is their joint metric computed as Val × Uni × Nov, representing the percentage of molecules that are simultaneously valid, unique, and novel. "ASta" and "MSta" denote atom-level and molecule-level stability. "Top" indicates the proportion of generated molecules that simultaneously satisfy all three property constraints, using relaxed cutoffs (QED > 0.4, SAS < 8, and binding affinity < –4.5) to avoid missing potentially useful candidates. All metrics are reported as percentages, and higher values indicate better performance.

constraints. These results highlight the robustness and effectiveness of our framework in optimizing multiple molecular properties.

### 4.2.3 Ablation Study

We conducted ablation studies to assess the contribution of each component in our framework (Table 2). Compared to traditional multi-objective strategies as well as other alternative uncertainty-based approaches, our method consistently outperforms across key metrics. While uniqueness slightly decreases, the trade-off is acceptable given the substantial gains in other objectives. Simplified variants of our method show clear performance drops, confirming the necessity of the full design.

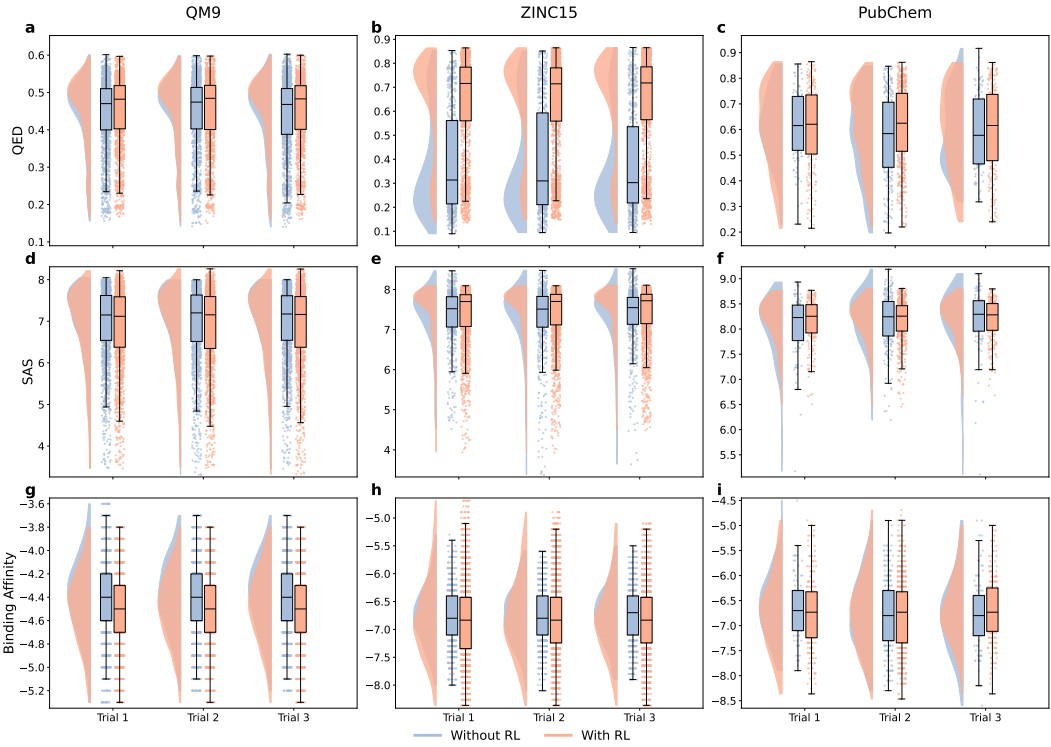

Figure 3: **Distributions of generated valid molecule properties across three datasets with and without RL-guided optimization.** (a–c) QED distributions for vanilla diffusion models pre-trained on QM9, ZINC15, and PubChem. (d–f) SAS distributions. (g–i) Binding affinity distributions. Each dataset shows results from three independent trials comparing vanilla diffusion models (without RL) and RL-guided diffusion models (with RL).

### 4.2.4 MD and ADMET analysis

We further evaluated the drug-likeness and stability of top-ranked molecules generated by our RL-guided diffusion models. As shown in Fig. 4, these candidates were compared with known EGFR mutant inhibitors using molecular docking, MD simulations, and ADMET profiling. The reference inhibitors showed stable RMSD trajectories and favorable pharmacokinetic profiles (Figs. 4e–f). Based on ADMET and MD performance, we found all candidates (molecules in Figs. 4b–d) maintained stable protein-ligand complexes, with RMSD values within or below reference ranges. Notably, candidates in Fig. 4c and Fig. 4d showed particularly strong conformational stability. ADMET analysis further confirmed good absorption, low CYP inhibition, and minimal toxicity. Overall, the generated compounds matched or outperformed the reference inhibitors, demonstrating the practical utility of our framework in identifying structurally stable, pharmacologically promising molecules for early-stage drug discovery.

## 5 Conclusion

We proposed an uncertainty-aware multi-objective RL-guided diffusion model framework for 3D de novo molecular generation. By incorporating predictive uncertainty into optimization, our method effectively balances complex and conflicting molecular design objectives. Experiments across multiple datasets show that RL-guided diffusion models generate high-quality molecules with superior validity, uniqueness, novelty, stability, and multiple drug-relevant properties. Notably, our method identified novel candidates with strong drug-like characteristics, MD stability, and favorable ADMET properties, beyond the reach of standard diffusion models. These results underscore the potential of our framework for early-stage drug discovery.

Table 2: **Ablation analysis results.**

| Category | Method | Val (%) (↑) | Uni (%) (↑) | Nov (%) (↑) | VUN (%) (↑) | ASta (%) (↑) | MSta (%) (↑) | Top (%) (↑) |
|---|---|---|---|---|---|---|---|---|
| Scalarization-based | WS | 91.78 ± 0.45 | 95.92 ± 0.84 | 99.81 ± 0.10 | 87.86 ± 0.81 | 99.52 ± 0.14 | 96.80 ± 0.74 | 27.02 ± 0.88 |
| | POO | 89.13 ± 0.54 | 87.86 ± 1.15 | 99.45 ± 0.07 | 77.88 ± 1.43 | 98.57 ± 0.17 | 91.27 ± 0.96 | 24.60 ± 1.61 |
| | MMM | 90.08 ± 0.80 | 79.55 ± 1.29 | 99.35 ± 0.05 | 71.20 ± 1.80 | 98.56 ± 0.19 | 91.33 ± 0.86 | 26.53 ± 2.09 |
| | LSDW | 91.37 ± 0.07 | 76.96 ± 1.06 | 99.48 ± 0.05 | 69.95 ± 1.00 | 99.27 ± 0.07 | 94.73 ± 0.38 | 21.73 ± 1.71 |
| Constraint-based | NMD | 93.30 ± 0.45 | 83.57 ± 1.33 | 99.61 ± 0.08 | 77.67 ± 1.27 | 99.45 ± 0.03 | 96.08 ± 0.31 | 25.75 ± 1.30 |
| | NMD-WS | 92.88 ± 0.07 | 76.71 ± 1.53 | 99.39 ± 0.10 | 70.82 ± 1.50 | 99.37 ± 0.09 | 95.20 ± 0.60 | 23.15 ± 1.36 |
| | CP | 92.15 ± 0.29 | 85.19 ± 0.99 | 99.49 ± 0.13 | 78.10 ± 1.06 | 99.09 ± 0.16 | 94.20 ± 1.10 | 26.30 ± 1.23 |
| | PFM | 91.98 ± 0.60 | 96.70 ± 0.40 | 99.77 ± 0.17 | 88.75 ± 0.78 | 99.61 ± 0.04 | 97.58 ± 0.28 | 24.68 ± 1.16 |
| Gradient-based | SVD | 84.62 ± 1.48 | 95.34 ± 1.24 | 99.57 ± 0.25 | 80.32 ± 0.74 | 97.72 ± 0.09 | 87.70 ± 0.32 | 25.62 ± 0.25 |
| | PCGrad | 86.98 ± 0.45 | 92.60 ± 0.53 | 99.46 ± 0.27 | 80.12 ± 0.87 | 98.23 ± 0.25 | 90.18 ± 0.62 | 24.87 ± 0.33 |
| | CAGrad | 86.33 ± 0.95 | 93.86 ± 0.40 | 99.61 ± 0.15 | 80.72 ± 0.78 | 98.30 ± 0.15 | 90.35 ± 0.74 | 24.27 ± 0.90 |
| | GradVac | 88.50 ± 0.97 | 96.16 ± 0.40 | 99.69 ± 0.07 | 84.83 ± 0.83 | 99.15 ± 0.08 | 90.45 ± 0.45 | 24.43 ± 0.26 |
| Uncertainty-based | UCB | 86.10 ± 0.95 | 95.59 ± 0.48 | **99.98 ± 0.04** | 82.28 ± 0.48 | 99.51 ± 0.08 | 97.12 ± 0.55 | 13.40 ± 0.20 |
| | EI | 85.68 ± 0.95 | 92.61 ± 0.46 | 99.50 ± 0.14 | 78.95 ± 0.86 | 98.00 ± 0.11 | 88.88 ± 0.62 | 26.97 ± 0.72 |
| | MVC | 87.78 ± 0.34 | **97.23 ± 0.51** | 99.80 ± 0.15 | 85.18 ± 0.09 | 99.03 ± 0.02 | 94.42 ± 0.28 | 24.03 ± 0.62 |
| | BORE | 89.33 ± 0.77 | 97.09 ± 0.35 | 99.81 ± 0.15 | 86.57 ± 1.14 | 99.42 ± 0.04 | 96.40 ± 0.44 | 23.73 ± 1.62 |
| Ours W/O | Reward Boost | 90.00 ± 0.52 | 96.93 ± 0.69 | 99.68 ± 0.08 | 86.95 ± 0.17 | 99.22 ± 0.03 | 95.40 ± 0.26 | 25.92 ± 1.54 |
| | Diversity Penalty | 83.55 ± 1.39 | 79.37 ± 0.76 | 99.17 ± 0.02 | 65.77 ± 1.40 | 96.17 ± 0.34 | 80.42 ± 1.77 | 25.43 ± 0.71 |
| | Static Cutoff | 95.73 ± 0.57 | 94.76 ± 0.28 | 99.93 ± 0.04 | 90.65 ± 0.76 | 99.87 ± 0.04 | 99.03 ± 0.28 | 24.88 ± 0.90 |
| | **Ours[†]** | **98.17 ± 0.07** | 90.90 ± 0.72 | 99.63 ± 0.04 | **88.90 ± 0.68** | **99.87 ± 0.03** | **99.17 ± 0.27** | **28.33 ± 0.61** |

[†] This repeats previous results for easier comparison.

Note: Each model generates 2,000 molecules per run. Results are averaged over three independent runs and reported as mean ± 95% confidence interval. All experiments in this table are conducted on the diffusion models pre-trained on QM9. "Val", "Uni", and "Nov" represent the percentages of valid, unique, and novel molecules, respectively. "VUN" is their joint metric computed as Val × Uni × Nov, representing the percentage of molecules that are simultaneously valid, unique, and novel. "ASta" and "MSta" denote atom-level and molecule-level stability. "Top" indicates the proportion of generated molecules that simultaneously satisfy all three property constraints, using relaxed cutoffs (QED > 0.4, SAS < 8, and binding affinity < –4.5) to avoid missing potentially useful candidates. "Ours W/O" refers to our framework with one module (reward boost, diversity penalty, or dynamic cutoff) ablated. All metrics are reported as percentages, and higher values indicate better performance.

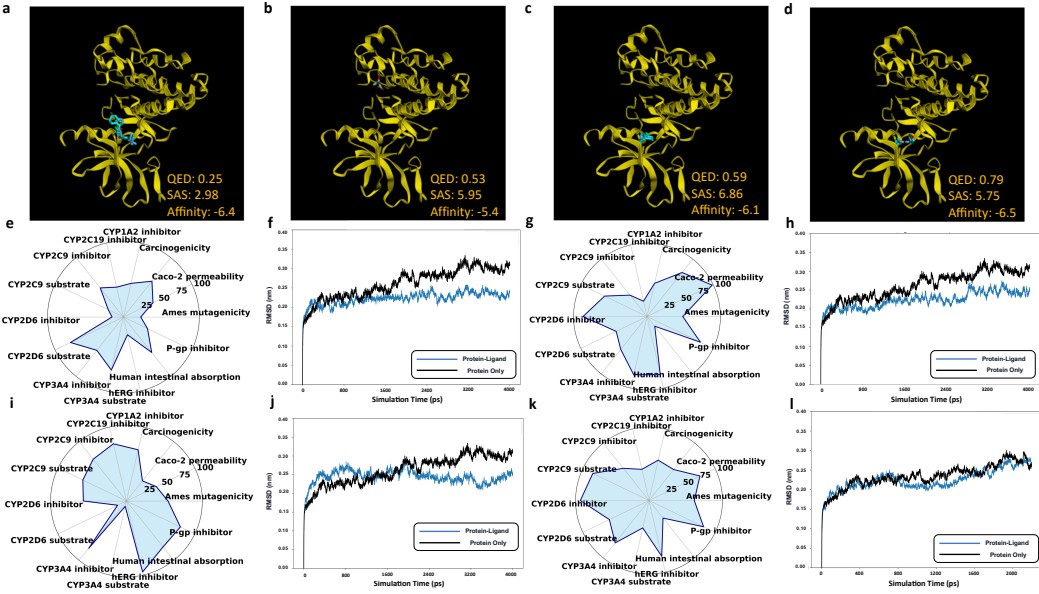

Figure 4: **Comparison between a known EGFR inhibitor and top molecules generated by RL-guided diffusion models.** (a) Docking pose of a known EGFR inhibitor. (b–d) Docking poses of the top-ranked molecules generated by RL-guided diffusion models pre-trained on QM9, ZINC15, and PubChem, respectively. QED, SAS, and binding affinity values are shown in each panel. (e–l) Corresponding ADMET radar plots and MD simulation RMSD curves. (e, f) Known inhibitor. (g, h) Molecule from (c). (i, j) Molecule from (b). (k, l) Molecule from (d). All MD trajectories were verified to have reached equilibrium prior to analysis, as indicated by the stabilization of RMSD values within a stable range of approximately 0.20–0.30 nm after an initial relaxation phase and the absence of sustained drift or abrupt fluctuations. The presented molecules were selected based on their overall performance.

**Limitations.** While our approach shows strong overall performance, results on PubChem suggest that existing diffusion architectures may struggle with large, complex molecules containing many heavy atoms. This limitation stems from backbone scalability rather than the RL framework itself. Future work on scalable diffusion architectures may further improve performance on large-molecule datasets.

**Broader Impact.** This work presents a generalizable framework that may accelerate early-stage drug discovery by efficiently exploring chemical space and optimizing multiple pharmacologically relevant properties. It enables the targeted design of molecules that meet predefined therapeutic criteria. Beyond drug discovery, the approach may benefit materials science, catalyst design, and molecular engineering. We acknowledge the ethical implications of powerful generative tools and support their responsible use within appropriate regulatory and safety frameworks.

## ACKNOWLEDGMENTS

This work was supported in part by the Canada Research Chairs Tier II Program (CRC-2021-00482), the Canadian Institutes of Health Research (PLL 185683, PJT 190272, PJT204042), the Natural Sciences, Engineering Research Council of Canada (RGPIN-2021-04072) and The Canada Foundation for Innovation (CFI) John R. Evans Leaders Fund (JELF) program (#43481), the Studentship/Fellowship funded by Breast Cancer Canada, and the Vector Scholarship in Artificial Intelligence provided through the Vector Institute.

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

# Appendix

## A   Methodology

This section provides extended details to supplement Section 3 of the main manuscript.

### A.1   Conditional Diffusion Model

**Notation.** We define $\mathbf{x} \in \mathbb{R}^{M \times 3}$ as the coordinates of $M$ atoms in 3D space, constrained to the zero center of gravity subspace, meaning the sum of the coordinates $\sum_i \mathbf{x}_i = 0$. The atom features, $\mathbf{h} \in \mathbb{R}^{M \times d}$, are invariant to E(3) transformations and include attributes like one-hot encoded atom types or charges. The conditioning variable $\mathbf{c}$ represents a desired molecular property. At each diffusion step $t$, the latent variable $\mathbf{z}_t = \left[ \mathbf{z}_t^{(x)}, \mathbf{z}_t^{(h)} \right]$ combines the noised coordinates $\mathbf{z}_t^{(x)}$ and features $\mathbf{z}_t^{(h)}$. The parameters $\alpha_t$ and $\sigma_t$ control the balance between signal retention and noise addition, while $\phi(\cdot, t, \mathbf{c})$ is an Equivariant Graph Neural Network (EGNN) that predicts noise during denoising, conditioned on both the time step $t$ and the property $\mathbf{c}$. Finally, $\mathcal{N}_{xh}$ denotes a joint normal distribution over coordinates and features, with coordinates constrained to the zero center of gravity subspace.

**Forward Diffusion.**   The forward diffusion process is the first key component of EDM, designed to gradually corrupt the original data into noise, creating a sequence of latent variables $\mathbf{z}_0, \mathbf{z}_1, \ldots, \mathbf{z}_T$ that we can later denoise to generate new samples. In a standard diffusion model, the process adds noise to a data point $\mathbf{x}$ according to the formula:

$$q(\mathbf{z}_t \mid \mathbf{x}) = \mathcal{N}(\mathbf{z}_t; \alpha_t \mathbf{x}, \sigma_t^2 \mathbf{I}), \tag{A.1}$$

where $\alpha_t$ determines how much of the original signal is retained, and $\sigma_t^2$ controls the variance of the added Gaussian noise. This formula captures the essence of diffusion: as $t$ increases, the data is progressively noised until it resembles pure noise at $t = T$.

In EDM, this idea is extended to jointly model both coordinates $\mathbf{x}$ and features $\mathbf{h}$, since molecules require both positional and categorical information. The forward process is Markovian, meaning each step depends only on the previous one, and is expressed as:

$$q(\mathbf{z}_0, \mathbf{z}_1, \ldots, \mathbf{z}_T \mid \mathbf{x}, \mathbf{h}) = q(\mathbf{z}_0 \mid \mathbf{x}, \mathbf{h}) \prod_{t=1}^{T} q(\mathbf{z}_t \mid \mathbf{z}_{t-1}). \tag{A.2}$$

The initial distribution at $t = 0$ is defined as:

$$q(\mathbf{z}_0 \mid \mathbf{x}, \mathbf{h}) = \mathcal{N}_{xh}(\mathbf{z}_0 \mid \alpha_0[\mathbf{x}, \mathbf{h}], \sigma_0^2 \mathbf{I}), \tag{A.3}$$

and the transition from $\mathbf{z}_{t-1}$ to $\mathbf{z}_t$ is given by:

$$q(\mathbf{z}_t \mid \mathbf{z}_{t-1}) = \mathcal{N}_{xh}(\mathbf{z}_t \mid \alpha_{t|t-1} \mathbf{z}_{t-1}, \sigma_{t|t-1}^2 \mathbf{I}), \tag{A.4}$$

where $\alpha_{t|t-1} = \alpha_t / \alpha_{t-1}$ adjusts the signal scaling between steps, and the noise variance is

$$\sigma_{t|t-1}^2 = \sigma_t^2 - \alpha_{t|t-1}^2 \sigma_{t-1}^2.$$

The joint distribution $\mathcal{N}_{xh}$ factorizes into distributions for coordinates and features:

$$\mathcal{N}_{xh}(\mathbf{z}_t \mid \alpha_t[\mathbf{x}, \mathbf{h}], \sigma_t^2 \mathbf{I}) = \mathcal{N}_x(\mathbf{z}_t^{(x)} \mid \alpha_t \mathbf{x}, \sigma_t^2 \mathbf{I}) \cdot \mathcal{N}(\mathbf{z}_t^{(h)} \mid \alpha_t \mathbf{h}, \sigma_t^2 \mathbf{I}). \tag{A.5}$$

Here, $\mathcal{N}_x$ is a normal distribution over the zero center of gravity subspace, ensuring the coordinates remain translation-invariant, and is defined as:

$$\mathcal{N}_x(\mathbf{x} \mid \boldsymbol{\mu}, \sigma^2 \mathbf{I}) = (\sqrt{2\pi}\sigma)^{-(M-1) \cdot 3} \exp\left( -\|\mathbf{x} - \boldsymbol{\mu}\|^2 / (2\sigma^2) \right), \tag{A.6}$$

with $\boldsymbol{\mu}$ in the zero center of gravity subspace. The feature distribution $\mathcal{N}$ is a standard normal distribution, reflecting the invariance of $\mathbf{h}$ to E(3) transformations.

The noise schedule is designed to be variance-preserving, meaning $\alpha_t = \sqrt{1 - \sigma_t^2}$, ensuring that the total variance of the data remains constant as noise is added. The specific form of $\alpha_t$ is chosen to smoothly transition from retaining the signal to adding noise:

$$\alpha_t = (1 - 2s) \cdot \left(1 - (t/T)^2\right) + s, \tag{A.7}$$

where $s = 10^{-5}$ is a small constant to prevent numerical issues, allowing $\alpha_t$ to decrease from $\alpha_0 \approx 1$ (almost pure signal) to $\alpha_T \approx 0$ (almost pure noise). To quantify the balance between signal and noise at each step, we define the Signal-to-Noise Ratio (SNR):

$$\text{SNR}(t) = \alpha_t^2 / \sigma_t^2 = \alpha_t^2 / (1 - \alpha_t^2). \tag{A.8}$$

The SNR decreases as $t$ increases, reflecting the increasing dominance of noise. For computational convenience during training and sampling, we also compute the negative log-SNR, defined as:

$$\gamma(t) = -\log \text{SNR}(t) = -\log \alpha_t^2 + \log \sigma_t^2. \tag{A.9}$$

The purpose of the negative log-SNR is to provide a numerically stable representation of the noise level at each step, which is particularly useful when optimizing the model or scheduling the noise. Using properties of the sigmoid function, we can express $\alpha_t^2 = \text{sigmoid}(-\gamma(t))$ and $\sigma_t^2 = \text{sigmoid}(\gamma(t))$, which simplifies calculations and ensures that the noise schedule is well-behaved across all timesteps.

**Backward Diffusion.** The goal of the reverse process is to generate new molecules by denoising the latent variables, starting from pure noise at $t = T$ and iteratively refining them back to a data sample at $t = 0$. In a conditional diffusion model, the reverse process approximates the denoising step, conditioned on $\mathbf{c}$, as:

$$p(\mathbf{z}_{t-1} \mid \mathbf{z}_t, \mathbf{c}) = \mathcal{N}(\mathbf{z}_{t-1}; \mu_\theta(\mathbf{z}_t, t, \mathbf{c}), \sigma_t^2 \mathbf{I}), \tag{A.10}$$

where $\mu_\theta(\mathbf{z}_t, t, \mathbf{c})$ is the mean predicted by EGNN, guiding the denoising process toward samples that satisfy the condition $\mathbf{c}$. This formula provides a high-level view of the reverse process, capturing how the model learns to reverse the noise addition step by step.

We define this process more precisely to ensure E(3) equivariance. The true posterior for the reverse step is:

$$q(\mathbf{z}_s \mid \mathbf{x}, \mathbf{h}, \mathbf{z}_t) = \mathcal{N}_{xh}(\mathbf{z}_s \mid \mu_{t \to s}([\mathbf{x}, \mathbf{h}], \mathbf{z}_t), \sigma_{t \to s}^2 \mathbf{I}), \tag{A.11}$$

where $s < t$, and the mean and variance are:

$$\mu_{t \to s}([\mathbf{x}, \mathbf{h}], \mathbf{z}_t) = (\alpha_{t|s} \sigma_s^2 / \sigma_t^2) \mathbf{z}_t + (\alpha_s \sigma_{t|s}^2 / \sigma_t^2)[\mathbf{x}, \mathbf{h}], \tag{A.12}$$

$$\sigma_{t \to s} = (\sigma_{t|s} \sigma_s) / \sigma_t, \quad \alpha_{t|s} = \alpha_t / \alpha_s, \quad \sigma_{t|s}^2 = \sigma_t^2 - \alpha_{t|s}^2 \sigma_s^2. \tag{A.13}$$

Since $\mathbf{x}$ and $\mathbf{h}$ are unknown during generation, we approximate them using EGNN $\phi(\mathbf{z}_t, t, \mathbf{c})$, which predicts the noise $\hat{\boldsymbol{\epsilon}}_t = [\hat{\boldsymbol{\epsilon}}_t^{(x)}, \hat{\boldsymbol{\epsilon}}_t^{(h)}]$. The estimated data is then computed as:

$$[\hat{\mathbf{x}}, \hat{\mathbf{h}}] = (\mathbf{z}_t / \alpha_t) - (\sigma_t / \alpha_t) \hat{\boldsymbol{\epsilon}}_t, \tag{A.14}$$

where $\hat{\boldsymbol{\epsilon}}_t = \phi(\mathbf{z}_t, t, \mathbf{c})$. Substituting this into the mean, the reverse transition distribution becomes:

$$p(\mathbf{z}_s \mid \mathbf{z}_t, \mathbf{c}) = \mathcal{N}_{xh}(\mathbf{z}_s \mid \mu_{t \to s}([\hat{\mathbf{x}}, \hat{\mathbf{h}}], \mathbf{z}_t), \sigma_{t \to s}^2 \mathbf{I}). \tag{A.15}$$

For sampling, we start with $\mathbf{z}_T \sim \mathcal{N}_{xh}(0, \mathbf{I})$, representing pure noise, and iteratively apply the reverse process for $t = T$ down to $t = 1$, setting $s = t - 1$. At each step, we sample $\boldsymbol{\epsilon} \sim \mathcal{N}(0, \mathbf{I})$, subtract the center of gravity from $\mathbf{z}_t^{(x)}$, compute the predicted noise $\hat{\boldsymbol{\epsilon}}_t = \phi(\mathbf{z}_t, t, \mathbf{c})$, and update:

$$\mathbf{z}_s = (1/\alpha_{t|s}) \mathbf{z}_t - (\sigma_{t|s}^2 / (\alpha_{t|s} \sigma_t)) \hat{\boldsymbol{\epsilon}}_t + \sigma_{t \to s} \boldsymbol{\epsilon}. \tag{A.16}$$

Finally, we sample the data $(\mathbf{x}, \mathbf{h}) \sim p(\mathbf{x}, \mathbf{h} \mid \mathbf{z}_0, \mathbf{c})$. EGNN ensures that the predicted noise for coordinates is equivariant:

$$\mathbf{R} \hat{\boldsymbol{\epsilon}}_t^{(x)} = \phi^{(x)}(\mathbf{R} \mathbf{z}_t^{(x)}, \mathbf{z}_t^{(h)}, t, \mathbf{c}), \tag{A.17}$$

where $\mathbf{R}$ is an orthogonal matrix.

**Training Objective.** To train the model, we aim to maximize the conditional log-likelihood $\log p(\mathbf{x}, \mathbf{h} \mid \mathbf{c})$, which is achieved by optimizing a variational lower bound:

$$\log p(\mathbf{x}, \mathbf{h} \mid \mathbf{c}) \geq \mathcal{L}_{c,0} + \mathcal{L}_{c,\text{base}} + \sum_{t=1}^{T} \mathcal{L}_{c,t}. \tag{A.18}$$

The term $\mathcal{L}_{c,t} = -\text{KL}(q(\mathbf{z}_s \mid \mathbf{x}, \mathbf{h}, \mathbf{z}_t) \| p(\mathbf{z}_s \mid \mathbf{z}_t, \mathbf{c}))$ measures the divergence between the true and approximate reverse distributions for $t = 1, \ldots, T$. The term $\mathcal{L}_{c,0} = \log p(\mathbf{x}, \mathbf{h} \mid \mathbf{z}_0, \mathbf{c})$ evaluates the likelihood at the final step, and $\mathcal{L}_{c,\text{base}} = -\text{KL}(q(\mathbf{z}_T \mid \mathbf{x}, \mathbf{h}) \| p(\mathbf{z}_T))$ compares the forward process at $t = T$ to the prior. Since $\alpha_T \approx 0$, $\mathcal{L}_{c,\text{base}} \approx 0$.

For $t \geq 1$, the KL divergence simplifies to a noise prediction objective:

$$\mathcal{L}_{c,t} = \mathbb{E}_{\boldsymbol{\epsilon} \sim \mathcal{N}_{xh}(0,\mathbf{I})} \left[ (1/2)w(t)\|\boldsymbol{\epsilon}_t - \phi(\mathbf{z}_t, t, \mathbf{c})\|^2 \right], \tag{A.19}$$

where $w(t) = 1 - \text{SNR}(t-1)/\text{SNR}(t)$. Setting $w(t) = 1$ simplifies training and improves sample quality, reducing the objective to minimizing the mean-squared error between the true noise $\boldsymbol{\epsilon}_t$ and the predicted noise $\phi(\mathbf{z}_t, t, \mathbf{c})$. During training, we sample $t \sim \mathcal{U}(0, \ldots, T)$, noise $\boldsymbol{\epsilon} \sim \mathcal{N}(0, \mathbf{I})$, compute $\mathbf{z}_t = \alpha_t[\mathbf{x}, \mathbf{h}] + \sigma_t \boldsymbol{\epsilon}$, and minimize $\|\boldsymbol{\epsilon} - \phi(\mathbf{z}_t, t, \mathbf{c})\|^2$.

The zero-term $\mathcal{L}_{c,0}$ splits into contributions from coordinates and features: $\mathcal{L}_{c,0} = \mathcal{L}_{c,0}^{(x)} + \mathcal{L}_{c,0}^{(h)}$.

For coordinates, we approximate the posterior as:

$$q(\mathbf{x} \mid \mathbf{z}_0^{(x)}) \approx \mathcal{N}_x(\mathbf{x} \mid \mathbf{z}_0^{(x)}/\alpha_0, (\sigma_0^2/\alpha_0^2)\mathbf{I}), \tag{A.20}$$

and the generative distribution is:

$$p(\mathbf{x} \mid \mathbf{z}_0, \mathbf{c}) = \mathcal{N}(\mathbf{x} \mid (\mathbf{z}_0^{(x)}/\alpha_0) - (\sigma_0/\alpha_0)\boldsymbol{\epsilon}_0^{(x)}, (\sigma_0^2/\alpha_0^2)\mathbf{I}), \tag{A.21}$$

yielding:

$$\mathcal{L}_{c,0}^{(x)} = \mathbb{E}_{\boldsymbol{\epsilon}^{(x)} \sim \mathcal{N}_x(0,\mathbf{I})} \left[ \log Z - \frac{1}{2}\|\boldsymbol{\epsilon}^{(x)} - \phi^{(x)}(\mathbf{z}_0, 0, \mathbf{c})\|^2 \right], \tag{A.22}$$

where $Z = (\sqrt{2\pi} \cdot \sigma_0/\alpha_0)^{(M-1)\cdot 3}$.

For categorical features, the noising is

$$q(\mathbf{z}_t^{(h)} \mid \mathbf{h}) = \mathcal{N}(\mathbf{z}_t^{(h)} \mid \alpha_t \mathbf{h}^{\text{onehot}}, \sigma_t^2 \mathbf{I}), \tag{A.23}$$

and the generative distribution is a categorical distribution, approximating $\mathcal{L}_{c,0}^{(h)} \approx 0$ for small $\sigma_0$.

**Conditional Generation Mechanism** The purpose of conditional generation is to produce molecules $(\mathbf{x}, \mathbf{h}) \sim p(\mathbf{x}, \mathbf{h} \mid \mathbf{c}, M)$ that satisfy the desired property $\mathbf{c}$ and have $M$ atoms. We first sample $(\mathbf{c}, M) \sim p(\mathbf{c}, M)$ from a learned distribution estimated from the training data. Then, we sample pure noise $\mathbf{z}_T \sim \mathcal{N}_{xh}(0, \mathbf{I})$, iteratively denoise using $p(\mathbf{z}_{t-1} \mid \mathbf{z}_t, \mathbf{c})$, and finally sample the data $(\mathbf{x}, \mathbf{h}) \sim p(\mathbf{x}, \mathbf{h} \mid \mathbf{z}_0, \mathbf{c})$. The neural network $\phi(\mathbf{z}_t, t, \mathbf{c})$ incorporates $\mathbf{c}$ by concatenating it to the node features, guiding the denoising process toward the desired property.

## A.2 Surrogate Models for Multi-objective Uncertainty Quantification

**Uncertainty Modeling.** The uncertainty $U_{\text{single}}(m; \delta)$ encompasses both aleatoric and epistemic uncertainties, combined into the total variance

$$\sigma_{\text{total}}^2(m) = \sigma_a^2(m) + \sigma_e^2(m). \tag{A.24}$$

Aleatoric uncertainty $\sigma_a^2$ reflects inherent data noise, such as variability from computational simulations, while epistemic uncertainty $\sigma_e^2$ arises from model limitations due to limited training data.

In the Chemprop model with evidential learning, these are estimated using the Normal Inverse-Gamma (NIG) parameters:

$$\sigma_a^2 = \frac{\beta}{\alpha - 1}, \quad \text{and} \quad \sigma_e^2 = \frac{\beta}{\nu(\alpha - 1)}, \tag{A.25}$$

where $\beta$, $\alpha$, and $\nu$ are predicted by the model.

The Gaussian distribution assumption, with mean $\mu(m)$ and standard deviation

$$\sigma(m) = \sqrt{\sigma_{\text{total}}^2(m)}, \tag{A.26}$$

is reasonable because it allows the integral in $U_{\text{single}}(m; \delta)$ to compute a probability, validated by the model's calibration to real-world data variability.

**Uncertainty as a Threshold Probability.** The uncertainty $\sigma(m)$ directly informs the probability that a molecule's true property value exceeds the threshold $\delta$, computed as:

$$U_{\text{single}}(m; \delta) = \eta \int_{\delta}^{\infty} \frac{1}{\sigma(m)\sqrt{2\pi}} \exp\left(-\frac{1}{2}\left(\frac{x - \mu(m)}{\sigma(m)}\right)^2\right) dx. \tag{A.27}$$

This integral represents the cumulative probability under a Gaussian distribution, where a larger $\sigma(m)$ reduces the likelihood of exceeding $\delta$ due to increased spread, and a smaller $\sigma(m)$ increases it if $\mu(m)$ is favorable.

The inclusion of total uncertainty ($\sigma_{\text{total}}^2$) ensures that both aleatoric and epistemic contributions are considered, reflecting the overall reliability of the prediction.

**Multi-Objective Uncertainty.** The multi-objective uncertainty is defined as:

$$U_{\text{multi}}(m; \delta_1, \ldots, \delta_k) = \prod_{i=1}^{k} U_{\text{single}}(m; \delta_i). \tag{A.28}$$

This product is justified by assuming conditional independence of property uncertainties given the molecule, meaning the likelihood of meeting all thresholds $\delta_1, \ldots, \delta_k$ is the joint probability of individual successes. Each $U_{\text{single}}(m; \delta_i)$ measures the probability for one property, and the product ensures a molecule is rewarded only if it likely satisfies all objectives simultaneously.

This approach avoids trade-offs, penalizing molecules with low probability for any property, which is critical in RL where a balanced solution is needed, making it a coherent aggregation strategy.

## A.3 RL-guided Optimization

**Transition Probability.** We first employ the trained diffusion model to sample a set of molecules and record their diffusion trajectories. Each trajectory can be treated as a Markov chain. The transition probability between two states on the chain are defined in Equation (A.4). Its PDF format is given by

$$p_\theta(\mathbf{z}_{t-1} \mid \mathbf{z}_t, \mathbf{c}) = \frac{1}{(2\pi\sigma_t^2)^{d/2}} \exp\left(-\frac{1}{2\sigma_t^2}\|\mathbf{z}_{t-1} - \mu_\theta(\mathbf{z}_t, t, \mathbf{c})\|^2\right), \tag{A.29}$$

relying on the variance $\sigma_t^2$ and the latent variables $\mathbf{z}_t$ and $\mathbf{z}_{t-1}$. The variance $\sigma_t^2$ is computed based on the noise schedule in the diffusion process. Referring to Equation (A.13), we define

$$\sigma_{t \to s}^2 = \frac{\sigma_t^2}{\alpha_t} = \sigma_t^2 \frac{\sigma_s^2}{\alpha_s^2}, \tag{A.30}$$

where $\sigma_t^2 = 1 - \alpha_t$, ensuring that the variance increases as $t$ grows, reflecting the progressive addition of noise in the forward process.

The latent variable $\mathbf{z}_t$ is sampled iteratively during the backward process of the diffusion model. Starting from $\mathbf{z}_T \sim \mathcal{N}(0, \mathbf{I})$, each $\mathbf{z}_t$ for $t = T - 1, \ldots, 0$ is computed using Equation (A.16):

$$\mathbf{z}_s = \frac{1}{\sqrt{\alpha_{t|s}}}\mathbf{z}_t - \left(\frac{\sigma_{t \to s}^2}{\sqrt{\alpha_{t|s}}\sigma_t}\right)\hat{\epsilon} + \sigma_{t \to s}\epsilon, \tag{A.31}$$

where $\epsilon \sim \mathcal{N}(0, \mathbf{I})$ and $\hat{\epsilon} = \phi(\mathbf{z}_t, t, \mathbf{c})$. Subsequently, $\mathbf{z}_{t-1}$ is sampled from

$$\mathcal{N}(\mathbf{z}_{t-1} \mid \mu_{t \to s}([\mathbf{x}, \mathbf{h}], \mathbf{z}_t), \sigma_{t \to s}^2 \mathbf{I}), \tag{A.32}$$

ensuring the denoising trajectory aligns with the learned distribution.

**Reward Bonus Design.** The reward bonus $R_{\text{bonus}}(m)$ is determined by three binary flags: validity ($v_m$), uniqueness ($u_m$), and novelty ($n_m$), each taking values in $\{0, 1\}$. These flags indicate whether molecule $m$ satisfies the respective criteria. The bonus is formulated as a weighted linear combination:

$$R_{\text{bonus}}(m) = b_v v_m + b_u u_m + b_n n_m, \tag{A.33}$$

where $b_v$, $b_u$, and $b_n$ are positive scalars representing the reward contributions of validity, uniqueness, and novelty, respectively.

This reward-boosting design allows binary objectives related to generation quality to be integrated without inducing reward sparsity, thereby enabling unified optimization of both binary and continuous rewards in a multi-objective setting.

**Optimization.** We optimize the diffusion model using a PPO-style clipped surrogate loss. Gradients are accumulated over multiple sampled timesteps based on fixed trajectories, enabling stable updates under complex multi-objective rewards. The pseudocode is shown in Algorithm A.1.

---

**Algorithm A.1** RL-guided Optimization for Diffusion Models

---

1: Initialize diffusion model $\pi_\theta$, optimizer, and learning rate scheduler
2: **for** episode = 1 to $N$ **do**
3:     Freeze current policy as $\pi_{\theta_{\text{old}}}$
4:     Generate a batch of molecules using $\pi_{\theta_{\text{old}}}$
5:     Record complete diffusion trajectories and transition probabilities
6:     Estimate uncertainty and compute reward probabilities
7:     Measure molecular diversity, validity, uniqueness, and novelty
8:     Compute total rewards
9:     **for** each policy update over fixed trajectories **do**
10:         Sample timesteps $t_1, t_2, \ldots, t_K \sim \mathcal{U}(0, T)$
11:         **for** each sampled timestep $t$ **do**
12:             Evaluate log-probabilities under $\pi_\theta$ at timestep $t$
13:             Compute importance ratio $r(m) = \exp(\log p_\theta - \log p_{\theta_{\text{old}}})$
14:             Compute clipped surrogate loss
15:             Accumulate gradients
16:         **end for**
17:         Update $\pi_\theta$ using accumulated gradients
18:         Step learning rate scheduler
19:     **end for**
20: **end for**

---

# B    Experimental Settings

As illustrated in Fig. B.1, our pipeline consists of five key stages: data preprocessing, surrogate model training, diffusion model pretraining, RL-guided optimization of the diffusion model, and molecule generation with subsequent in silico evaluation. The following sections provide additional implementation details for each stage of the pipeline.

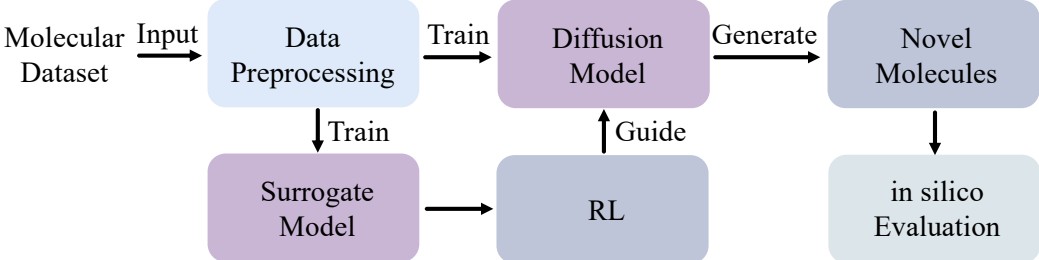

Figure B.1: **Overall workflow of our framework.** A molecular dataset is first preprocessed to train both the diffusion model and the surrogate models. RL guides the optimization of the diffusion model using predictions from the surrogate models. The optimized diffusion model then generates candidate molecules for in silico evaluation.

## B.1    Data Sources and Descriptions

We use three molecular datasets (QM9, ZINC15, and PubChem) as sources for training and evaluating surrogate and diffusion models. In addition, we use a curated crystallographic structure of mutant EGFR as the target receptor for molecular docking. Table B.1 summarizes the sources, descriptions, and download options for each dataset used in this study.

## B.2    Data Preprocessing

### B.2.1    Target Protein Preprocessing

We selected the human EGFR as the docking target in this study due to its central role in cell signaling and its clinical importance in various cancers. In particular, we used the 3D crystal structure with PDB ID 6VHN, which corresponds to the kinase domain of a mutant form of EGFR bound to a small-molecule inhibitor. This structure provides a biologically relevant conformation for evaluating ligand binding and offers sufficient resolution (2.5 Å) for structure-based modeling.

To prepare the protein for molecular docking, we applied a series of standard preprocessing steps to ensure structural completeness and compatibility with docking software. First, all water molecules, ions, and co-crystallized ligands not involved in the binding site were removed to eliminate potential artifacts and ensure a clean receptor surface. The structure was then processed to convert any non-standard residues into their canonical forms, and missing side chains were reconstructed using geometry-based modeling to ensure a complete atomic representation. Polar hydrogen atoms were added throughout the protein to restore potential hydrogen bonding interactions typically unresolved in crystallographic data. Finally, partial atomic charges were assigned to the protein using a standard method compatible with docking engines. The resulting structure was saved in the appropriate format and used as the receptor input in subsequent docking simulations.

### B.2.2    Molecular Data Preprocessing

To process each dataset, we first remove invalid molecules (e.g., those with missing 3D coordinates or broken structures) and eliminate duplicates. For the remaining molecules, we annotate three key properties (QED, SAS, and binding affinity) for each molecule. Specifically, QED and SAS are computed using RDKit, while the binding affinity is predicted using AutoDock Vina through docking simulations with the EGFR receptor site. These property annotations will be used in both surrogate model training and diffusion model training.

Table B.1: **Sources and descriptions of datasets.**

| Dataset | Item | Details |
|---|---|---|
| QM9 | Source | `http://deepchem.io.s3-website-us-west-1.amazonaws.com/` `datasets/gdb9.tar.gz` |
| | Description | Dataset containing quantum chemical properties of small organic molecules. QM9 is widely used in the development and evaluation of machine learning models for molecular property prediction. |
| | Download Option | Whole dataset (SDF format) |
| ZINC15 | Source | `https://zinc15.docking.org/tranches/home/#` |
| | Description | A free database of commercially available molecules for virtual screening, frequently used in drug discovery research. |
| | Download Option | 3D lead-like molecules (SDF format); neutral or +0 charged; mid-range logP; in-stock availability; standard reactivity |
| PubChem | Source | `https://pubchem.ncbi.nlm.nih.gov/#query=small%20molecule&` `tab=compound` |
| | Description | A public chemical database providing bioactivity data and annotations for small molecules. PubChem is extensively used in cheminformatics, drug discovery, and bioinformatics applications. |
| | Download Option | Small molecule subset (SDF format) |
| EGFR | Source | `https://www.rcsb.org/structure/6VHN` |
| | Description | Crystallographic structure of the kinase domain of mutant human EGFR (PDB ID: 6VHN), co-crystallized with a small-molecule inhibitor. This structure serves as the receptor target for binding affinity prediction via molecular docking. |
| | Download Option | Downloadable in .pdb format from RCSB PDB |

After property computation, all hydrogen atoms are removed from the 3D molecular structures. This decision is motivated by the need to reduce computational overhead and simplify molecular graphs during preprocessing and training. Hydrogen atoms offer limited topological information and can be accurately reconstructed at a later stage based on standard valency rules. Because the core chemical and spatial information of each molecule remains intact, this step introduces no adverse impact on the training performance. Each molecule is ultimately represented by its SMILES string, 3D structure (including atom types and coordinates), and its drug-related properties. The SMILES string is used as molecular input when training surrogate models, while the 3D structure is used as input to the diffusion model.

To train the diffusion model, we further construct dataset-level configuration files based on the hydrogen-removed molecules. For each dataset, each distinct atom type in the molecule is assigned a numerical encoding via an atom encoder and its corresponding decoder. The unique atom types identified in each dataset are listed in Table B.2. The number of atoms in each molecule is counted to determine the maximum graph size across the dataset. Fig B.2 illustrates the number of molecules in each dataset grouped by heavy atom count. Molecules in QM9 typically contain fewer than 10 heavy atoms, while those in ZINC15 and PubChem span a broader range of sizes. Especially in PubChem, some structures exceed 40 atoms.

Table B.2: **Unique atom types observed in each dataset.** (after hydrogen removal)

| Dataset | Unique Atom Types |
|---|---|
| QM9 | C, N, O, F |
| ZINC15 | C, N, O, F, P, S, Cl, Br, I |
| PubChem | C, N, O, F, P, S, Cl, Br, I |

To prepare the data for model training and evaluation, we apply different data splitting strategies tailored to the modeling task.

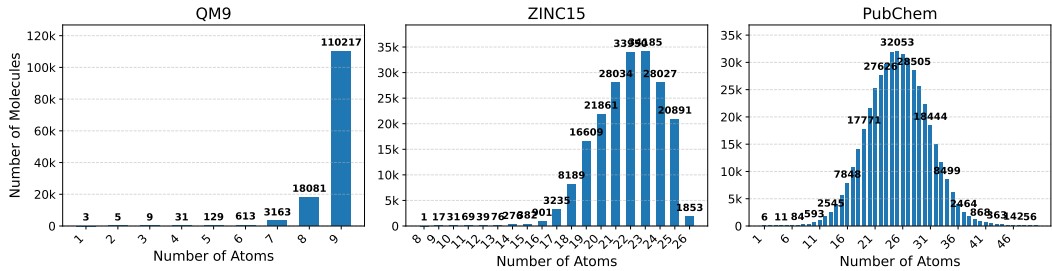

Figure B.2: **Distribution of molecules by the number of heavy atoms in each dataset.** Subfigures (a), (b), and (c) correspond to QM9, ZINC15, and PubChem, respectively.

**Scaffold-based splitting.** For the surrogate model, we adopt scaffold-based splitting. Specifically, each molecule is first assigned a Bemis–Murcko scaffold by removing its side chains and retaining the core ring and linker structures. Molecules sharing the same scaffold are grouped together. These scaffold groups are then randomly shuffled and assigned to the training, validation, and test sets in such a way that no scaffold appears in more than one set. This ensures that the model is evaluated on structurally novel scaffolds not seen during training.

**Species-based splitting.** For training the diffusion model, we use species-based splitting. Each molecule is grouped based on the unique set of atom types it contains (i.e., its chemical species). These species groups are shuffled and assigned to disjoint training, validation, and test partitions, ensuring that each subset contains distinct combinations of atomic species. This strategy preserves the global diversity of atom types while promoting generalization to structurally diverse molecules in 3D space.

Table B.3 summarizes the dataset statistics and splitting outcomes, including the total number of molecules, maximum number of heavy atoms, and the number of samples in the training, validation, and test sets for each dataset.

Table B.3: **Dataset statistics after preprocessing.** Hydrogen atoms are excluded in max atom count.

|  | ZINC15 | QM9 | PubChem |
|---|---|---|---|
| # total molecules | 198,626 | 132,251 | 444,287 |
| # max atoms | 26 | 9 | 50 |
| # molecules in train set (80%) | 158,900 | 105,800 | 355,429 |
| # molecules in test set (10%) | 19,862 | 13,225 | 44,428 |
| # molecules in validate set (10%) | 19,864 | 13,226 | 44,430 |

## B.3 Surrogate Model Training and Hyper-parameters

We train surrogate models using Chemprop, a graph neural network-based molecular property predictor. For each dataset, we use the available SMILES strings and their corresponding property labels (e.g., QED, SAS, or binding affinity) to train a separate model. The trained surrogate model estimates not only the predicted property value but also the uncertainty associated with whether the value exceeds a predefined threshold. Model hyperparameters are selected via grid search, as summarized in Table B.4. The best hyperparameter configurations selected for each dataset are reported in Table B.5.

Table B.4: **Hyperparameter search space for surrogate model training.**

| Hyperparameter | Search Range |
|---|---|
| activation | ReLU, SELU, GELU, Leaky ReLU |
| batch size | 32, 64, 128, 512, 1024 |
| dropout | 0.0, 0.15, 0.1, 0.2 |
| ensemble size | 1, 5, 10, 20, 30 |
| epochs | 5, 10, 20, 30, 40, 50, 60 |
| evidential regularization | 1e-3, 5e-3, 1e-2 |
| ffn_hidden_size | 200, 300, 400, 600 |
| final learning rate | 1e-5, 1e-4 |
| hidden size | 200, 300, 400, 600 |
| loss function | mse, evidential, mve |
| ffn_num_layers | 2, 3 |
| max learning rate | 1e-3, 1e-2 |
| initial learning rate | 1e-4, 1e-3 |

Table B.5: **Hyperparameters for surrogate models across datasets and prediction tasks.**

| Hyperparameter | QM9 | | | ZINC15 | | | PubChem | | |
|---|---|---|---|---|---|---|---|---|---|
| | QED | SAS | Affinity | QED | SAS | Affinity | QED | SAS | Affinity |
| activation | PReLU | PReLU | PReLU | PReLU | PReLU | PReLU | PReLU | PReLU | PReLU |
| batch size | 64 | 64 | 64 | 64 | 64 | 64 | 64 | 64 | 64 |
| dropout | 0 | 0.15 | 0 | 0 | 0 | 0 | 0 | 0.1 | 0 |
| ensemble size | 1 | 1 | 1 | 1 | 1 | 1 | 1 | 1 | 1 |
| epochs | 40 | 40 | 40 | 40 | 40 | 40 | 40 | 40 | 40 |
| evidential regularization | 5e-3 | 5e-3 | 5e-3 | 5e-3 | 5e-3 | 5e-3 | 1e-2 | 5e-3 | 5e-3 |
| ffn_hidden_size | 300 | 300 | 300 | 300 | 300 | 300 | 300 | 300 | 300 |
| final learning rate | 1e-5 | 1e-5 | 1e-5 | 1e-5 | 1e-5 | 1e-5 | 1e-5 | 1e-5 | 1e-5 |
| hidden size | 300 | 300 | 300 | 300 | 300 | 300 | 300 | 300 | 300 |
| loss function | evidential | evidential | evidential | evidential | evidential | evidential | evidential | evidential | evidential |
| ffn_num_layers | 2 | 2 | 2 | 2 | 2 | 2 | 2 | 2 | 2 |
| max learning rate | 3e-3 | 3e-3 | 3e-3 | 3e-3 | 3e-3 | 3e-3 | 3e-3 | 3e-3 | 3e-3 |
| initial learning rate | 1e-4 | 1e-4 | 1e-4 | 1e-4 | 1e-4 | 1e-4 | 1e-4 | 1e-4 | 1e-4 |

## B.4 Hyper-parameters for RL-guided Diffusion Model Training

We perform grid search over key hyperparameters for RL-guided diffusion model training, as summarized in Table B.6. The final selected configurations for RL-guided diffusion models pre-trained on each dataset (QM9, ZINC15, PubChem) are reported in Table B.7.

Table B.6: **Hyperparameter search space for RL-guided diffusion model training.**

| Hyperparameter | Search Range |
|---|---|
| valid_bonus | 0.05, 0.2, 0.3, 0.35 |
| unique_bonus | 0.05, 0.2, 0.3, 0.35 |
| novel_bonus | 0.05, 0.2, 0.3, 0.35 |
| clip_param | 1e-4, 2e-4, 3e-4 |
| learning rate | 1e-5, 2e-5, 5e-5 |
| reuse | 1, 2, 3, 5, 10 |
| n_samples | 32, 64, 128, 256 |

## B.5 Baseline Implementation

For each baseline, we follow the original framework design proposed in the corresponding papers. To adapt our multi-objective setting to these single-objective optimization frameworks, we convert the task into a binary reward formulation: a molecule receives a reward of 1 only if all target properties exceed their predefined cutoffs, and 0 otherwise. The cutoffs are dataset-specific and determined based on the top 10% of molecules ranked by each property, which provides a well-defined reward signal while mitigating the issue of extreme reward sparsity. For SFT-PG, we instead define the reward as the discrepancy between the property distribution of the generated molecules and an

Table B.7: **Final hyperparameter settings for RL-guided diffusion model training.**

| Hyperparameter | QM9 | ZINC15 | PubChem |
|---|---|---|---|
| valid_bonus | 0.2 | 0.2 | 0.05 |
| unique_bonus | 0.35 | 0.35 | 0.3 |
| novel_bonus | 0.05 | 0.05 | 0.05 |
| clip_param | 3e-4 | 3e-4 | 3e-4 |
| rl_lr | 1e-5 | 1e-5 | 1e-5 |
| reuse | 3 | 3 | 3 |
| n_samples | 128 | 128 | 64 |

ideal target distribution derived from the unconditional diffusion model. We adopt hyperparameter configurations similar to the original implementations, using a clipping parameter of 0.2, a learning rate of $1 \times 10^{-5}$, and generating 128 samples per policy update.

## B.6 Ablation Study Design

We compare our method against three classes of widely used multi-objective optimization paradigms (scalarization-based, constraint-based, and gradient-based) to validate that the performance gains of our uncertainty-aware reward are not simply due to reformulating the objectives. These methods represent common and well-established strategies in multi-objective tasks.

We also compare to other uncertainty-based methods to demonstrate that our formulation has more actionable signals than generic uncertainty sampling or exploration strategies.

Lastly, we ablate key components in our reward module such as reward boosting, diversity penalty, and dynamic cutoff adjustment, which helps to understand their individual contributions and assess how each module improves molecule quality, stability, or balance across objectives.

## B.7 MD simulations and ADMET property prediction

To evaluate the dynamic stability and pharmacokinetic viability of candidate molecules beyond static docking, we conducted MD simulations and ADMET property predictions as two complementary computational assays. The MD simulations were implemented using AmberTools for system setup and OpenMM for GPU-accelerated production runs. Protein-ligand complexes were constructed by first assigning GAFF atom types to ligands using Antechamber, while proteins were parameterized with the ff14SB force field. All complexes were solvated in a rectangular TIP3P water box with a minimum 10 Å buffer in each direction and neutralized using counterions. The system topology and coordinate files were generated via tleap, and the entire simulation was conducted under periodic boundary conditions.

Each system underwent initial energy minimization and equilibration prior to production runs. The equilibration protocol consisted of a 100 ps NVT heating phase and a 200 ps NPT equilibration phase, gradually increasing the temperature to 300 K using a Langevin thermostat. The production simulations were run under the NPT ensemble at 1 atm and 300 K using Langevin dynamics with a 2 fs timestep, totaling 1,000,000 steps (corresponding to a total physical simulation time of 4,000 ps). PME was used for long-range electrostatics with a 10 Å cutoff, and all hydrogen-containing bonds were constrained.

The primary readout of MD simulation was the RMSD of protein backbone atoms, calculated over time using the initial apo structure as a reference. RMSD trajectories were used to assess the stability and equilibration of each protein–ligand complex. All systems showed RMSD stabilization within a narrow range of approximately 0.20–0.30 nm after an initial relaxation phase, confirming that equilibrium was reached prior to subsequent analysis. Complexes demonstrating such early convergence, low fluctuation, and the absence of sharp deviations were considered structurally stable. This dynamic screening step ensures that promising compounds maintain consistent binding behavior and do not induce substantial conformational drift or instability under physiologically relevant simulation conditions.

In parallel, ADMET properties were evaluated using ADMET-AI, a graph neural network–based platform trained on diverse experimental datasets. While the tool generates predictions for 41 pharmacokinetic and toxicological endpoints, we focused our analysis on a curated subset of 14 key properties that are particularly relevant for early-stage drug development. These include critical parameters across all five ADMET domains, which are absorption (e.g., human intestinal absorption, Caco-2 permeability), distribution (e.g., volume of distribution, blood-brain barrier permeability), metabolism (e.g., CYP450 isoform inhibition for 3A4, 2D6, and 2C9), excretion (e.g., renal clearance), and toxicity (e.g., hERG inhibition, Ames mutagenicity, hepatotoxicity, and LD50).

The rationale for selecting these 14 properties stems from their strong empirical association with clinical trial outcomes and their routine use in pharmaceutical pipelines as early-stage go/no-go filters. Many of the remaining 27 endpoints, such as secondary CYP isoforms or advanced carcinogenicity markers, are either redundant with the selected subset or less predictive during the hit-to-lead phase. The 14-feature radar profile thus serves as a concise yet informative representation of a molecule's drug-likeness and safety, enabling practical compound triage without overfitting to noisy or weakly generalizable predictions.

Each compound's profile was evaluated both as an individual feature vector and as an integrated radar chart to identify potential liabilities. Compounds with strong absorption and distribution but poor metabolic or toxicity profiles were flagged for exclusion. Conversely, molecules with well-balanced and favorable values across all 14 categories were considered pharmacokinetically viable. This ADMET analysis complements the MD results by ensuring that structurally stable ligands also meet essential safety and efficacy criteria, thereby reinforcing their prioritization for further validation.

## B.8 Evaluation

### B.8.1 Evaluation Metrics for Surrogate Models

We adopt a widely used evaluation framework that includes both regression accuracy and uncertainty calibration metrics to assess the performance of surrogate models trained with uncertainty prediction.

**Regression Accuracy.** We use the coefficient of determination ($R^2$) to assess the accuracy of the surrogate model's predicted mean values. This metric quantifies how well the predicted outputs align with the ground-truth labels. It is defined as

$$R^2 = 1 - \frac{\sum_i (y_i - \hat{y}_i)^2}{\sum_i (y_i - \bar{y})^2}, \tag{B.1}$$

where $y_i$ and $\hat{y}_i$ denote the ground-truth and predicted values, respectively, and $\bar{y}$ is the mean of the ground-truth values. An $R^2$ score of 1 indicates perfect prediction, whereas a score of 0 suggests that the model performs no better than simply predicting the mean. Higher $R^2$ values therefore reflect stronger predictive performance and tighter alignment between the model's outputs and observed data.

**Uncertainty Calibration.** To quantify how well predicted uncertainties reflect actual confidence, we compute the AUCE. This metric compares predicted confidence levels to the empirical frequency with which the true values fall within corresponding confidence intervals. It is defined as

$$\text{AUCE} = \int_0^1 \left| \hat{P}(c) - c \right| dc, \tag{B.2}$$

where $c \in [0, 1]$ denotes the confidence level, and $\hat{P}(c)$ is the proportion of predictions whose ground-truth values fall within the predicted $c$-level confidence interval. Lower AUCE values indicate better calibration.

**Visualization.** To complement the numerical metrics, we employ two types of visualizations for model diagnostics:

- **Parity plots**: compare predicted values against ground-truth targets, with predictive uncertainty shown as color gradients. We additionally include histograms and residual error plots to assess potential bias, variance, and distributional shifts.

- **Calibration curves**: plot empirical coverage $\hat{P}(c)$ versus nominal confidence level $c$, allowing a visual assessment of how well the predicted uncertainties align with observed outcomes.

These visualizations provide intuitive insights into both regression accuracy and uncertainty calibration, and help identify systematic errors that may not be apparent from $R^2$ or AUCE alone.

### B.8.2 Evaluation Metrics for Diffusion Models With and Without Optimization

To assess the quality generated molecules, we evaluate them using six key metrics: validity, uniqueness, novelty, atom stability, molecular stability, and the proportion of top molecules. Let $\mathcal{M}_{\text{gen}}$ denote the set of molecules generated by the model, and $\mathcal{M}_{\text{train}}$ the set of training molecules.

**Validity.** Validity measures the fraction of generated molecules that can be parsed into chemically valid molecular structures. Specifically, we consider a molecule valid if it can be successfully reconstructed from its atomic coordinates and types, converted into an RDKit molecule object, and translated into a canonical SMILES string without triggering errors during bond inference, sanitization, or stereochemistry assignment. This process includes constructing an XYZ block from the predicted positions and atom types, inferring bonds using RDKit's internal heuristics, and applying standard sanitization and stereochemical processing. If these steps succeed without exceptions, the molecule is considered valid. Formally, the validity score is defined as:

$$\text{Validity} = \frac{|\{m \in \mathcal{M}_{\text{gen}} \mid m \text{ is valid}\}|}{|\mathcal{M}_{\text{gen}}|} \times 100\% \tag{B.3}$$

**Uniqueness.** Uniqueness denotes the proportion of valid molecules that are distinct from each other. It is computed by removing duplicates among all valid molecules based on their canonical SMILES representations. A higher uniqueness score indicates that the model is capable of generating a chemically diverse set of valid structures, rather than repeatedly producing similar or identical molecules. Conversely, a low uniqueness score may suggest issues such as overfitting to the training data or mode collapse, where the model fails to explore a wide range of chemical space. Formally, it is defined as:

$$\text{Uniqueness} = \frac{|\text{Unique}(\mathcal{M}_{\text{valid}})|}{|\mathcal{M}_{\text{valid}}|} \times 100\% \tag{B.4}$$

**Novelty.** Novelty quantifies the proportion of unique valid molecules generated by the model that do not appear in the training set. This comparison is performed using canonical SMILES strings to ensure consistency in molecular representation. A higher novelty score indicates that the model is capable of generating novel chemical structures beyond those it has seen during training, which is desirable in de novo molecular design. In contrast, a low novelty score may suggest that the model has overfit to the training data and is merely replicating known molecules. Formally, it is defined as:

$$\text{Novelty} = \frac{|\{m \in \text{Unique}(\mathcal{M}_{\text{valid}}) \mid m \notin \mathcal{M}_{\text{train}}\}|}{|\text{Unique}(\mathcal{M}_{\text{valid}})|} \times 100\% \tag{B.5}$$

**VUN.** VUN measures the proportion of generated molecules that are simultaneously valid, unique, and novel. It is computed as the product of Validity, Uniqueness, and Novelty. This metric reflects the overall ability of the model to generate truly de novo compounds that are chemically correct, structurally diverse, and absent from the training set. A high VUN indicates that a large fraction of the generated molecules can be considered meaningful de novo candidates. In contrast, a low VUN suggests that only a small portion of outputs are genuinely novel and chemically usable, limiting the model's utility in de novo molecular design. Formally,

$$\text{VUN} = \text{Validity} \times \text{Uniqueness} \times \text{Novelty} = \frac{|\{m \in \text{Unique}(\mathcal{M}_{\text{valid}}) \setminus \mathcal{M}_{\text{train}}\}|}{|\mathcal{M}_{\text{gen}}|} \times 100\% \tag{B.6}$$

**Atom Stability.** Atom Stability measures the proportion of atoms in valid molecules that satisfy known valency rules. For each generated molecule, atom types and their 3D coordinates are used to reconstruct local bonding environments. Each atom is then evaluated based on whether its inferred number of bonds falls within its chemically allowed valence range. This metric captures whether the

model respects fundamental atom-level chemical constraints. A high atom stability score indicates that most atoms are placed in chemically reasonable configurations with proper bonding patterns. Formally, the metric is defined as:

$$\text{Atom Stability} = \frac{|\{m \in \mathcal{M}_{\text{valid}} \mid \text{all atoms in } m \text{ obey valency}\}|}{|\mathcal{M}_{\text{valid}}|} \times 100\% \qquad \text{(B.7)}$$

**Molecular Stability.** Molecular Stability measures the proportion of valid molecules that form chemically coherent structures as a whole. A molecule is considered stable if it forms a connected graph without unrealistic fragments, over-bonded atoms, or invalid substructures. Stability is assessed by reconstructing the full molecule from predicted positions and atom types, followed by heuristic bond inference and consistency checks. This metric reflects the model's ability to generate molecules that are not only locally plausible (per atom), but also structurally sound on a global level. A high molecular stability score suggests that the majority of generated molecules form valid chemical graphs without structural discontinuities or violations of bonding rules. Formally, the metric is defined as:

$$\text{Molecular Stability} = \frac{|\{m \in \mathcal{M}_{\text{valid}} \mid m \text{ is chemically stable}\}|}{|\mathcal{M}_{\text{valid}}|} \times 100\% \qquad \text{(B.8)}$$

**Top Molecules.** This metric calculates the percentage of generated molecules that simultaneously satisfy all three essential drug-relevant property constraints: QED $\geq 0.4$, SAS $\leq 8.0$, and binding affinity $\leq -4.5$ kcal/mol (QED and SAS scores are computed using RDKit, while binding affinity is estimated using AutoDock Vina). Table B.8 describes the meaning of different value ranges for QED, SAS, and binding affinity in the context of drug design. As a composite metric, it reflects the model's capacity to generate candidate compounds that are not only chemically valid but also pharmacologically meaningful and synthetically accessible. We require a minimum QED score of 0.4 to ensure baseline drug-likeness. In practice, molecules with higher QED scores are more likely to exhibit favorable physicochemical properties, including appropriate lipophilicity, molecular weight, and hydrogen-bonding potential. These characteristics are typically associated with improved pharmacokinetics, reduced toxicity, and enhanced bioavailability in vivo. We also apply a SAS threshold of 8.0 to filter out molecules that are prohibitively difficult to synthesize. This constraint reflects real-world development considerations, where compounds that require low-yield or complex synthetic routes are often deemed infeasible regardless of their predicted activity. Finally, we impose a binding affinity threshold of –4.5 kcal/mol to retain candidates that are thermodynamically capable of engaging the target protein with meaningful strength. For kinase targets like EGFR, effective binding at the Adenosine TriPhosphate (ATP) pocket is a prerequisite for competitive inhibition and downstream signaling blockade. Docking scores are inherently approximate. However, more negative

Table B.8: **Reference ranges and interpretability of QED, SAS, and binding affinity scores.**

| Property | Range | Interpretation |
|---|---|---|
| QED | (0.0, 0.3) | Low drug-likeness; poor pharmaceutical potential |
| | (0.3, 0.5) | Suboptimal but potentially useful structures |
| | (0.5, 0.7) | Typical lead-like drug candidates |
| | (0.7, 0.9) | High drug-likeness, good design quality |
| | (0.9, 1.0) | Excellent drug-likeness (rare) |
| SAS | (1.0, 3.0) | Very easy to synthesize; trivial structures |
| | (3.0, 5.0) | Easy to synthesize using standard routes |
| | (5.0, 6.5) | Moderately challenging to synthesize |
| | (6.5, 8.0) | Synthesis may require complex procedures and conditions |
| | (8.0, 10.0) | Extremely difficult or infeasible to synthesize |
| Binding Affinity (kcal/mol) | (-15.0, -9.0) | Very high binding affinity; near irreversible |
| | (-9.0, -7.0) | Strong binding; ideal for inhibitors |
| | (-7.0, -5.5) | Moderate binding; potentially active |
| | (-5.5, -4.5) | Binding is weak; potential for initial target engagement |
| | (-4.5, 0.0) | Very weak binding; typically inactive |

binding energies often correlate with stronger interactions and, by extension, higher inhibitory potential. To avoid prematurely discarding promising compounds, particularly during the exploratory phase of molecular screening, we adopt this permissive but principled filtering strategy that balances chemical diversity with essential real-world constraints. By requiring all three properties to be satisfied simultaneously, the metric avoids overestimating molecules that perform well in only one aspect while failing in others. At the same time, it ensures that slightly suboptimal values in one property do not automatically disqualify a molecule, as long as the overall profile remains within a plausible range. This joint criterion enables a more holistic assessment of generated molecules and better reflects the multidimensional requirements of early-stage drug-like candidates. Formally, the Top Molecules score is defined as follows:

$$\text{Top Molecules} = \frac{|\{m \in \mathcal{M}_{\text{novel}} \mid \text{QED}(m) \geq 0.4 \wedge \text{SAS}(m) \leq 8 \wedge \text{Affinity}(m) \leq -4.5\}|}{|\mathcal{M}_{\text{gen}}|} \times 100\%$$

(B.9)

## B.9   Devices and Computational Setup

All experiments are conducted on NVIDIA A100 GPUs with 80 GB of memory. On the QM9 dataset, both EDM and GeoLDM require approximately 90 seconds per training iteration during pretraining. Due to its additional modules and guidance mechanisms, GFMDiff takes around 5 minutes per iteration. During RL-based EDM optimization, which includes molecule generation, property evaluation, and sample reuse, each optimization step takes about 2 minutes. Generating a single molecule using EDM model requires approximately 20 seconds. On larger datasets, the training time increases accordingly. On ZINC15, one pretraining iteration of EDM takes around 7 minutes, while on PubChem it takes approximately 10 minutes. During optimization based on pretrained EDM models, each optimization step takes about 3 minutes on ZINC15 and 5 minutes on PubChem.

# C    Results and Discussion

## C.1    Correlation Analysis

The independence assumption in Equation 3 allows us to represent the multi-objective reward as a product of individual property-specific probabilities, where each probability corresponds to the likelihood that a generated molecule exceeds a predefined cutoff for a given property. These probabilities are estimated using uncertainty-aware predictions from Chemprop, and their product provides a smooth and differentiable reward signal suitable for RL.

In our study, we focus on three key properties: QED, SAS, and binding affinity. These properties are designed to reflect different and complementary aspects of molecule quality. Empirically, we find that these properties are only weakly correlated. As the correlation matrices are shown in Table C.1, all pairwise Pearson correlation coefficients between the properties are less than 0.3 in magnitude, indicating low correlation and supporting the independence assumption in this context.

Table C.1: **Correlation matrices of QED, SAS, and binding affinity for different datasets**

| QM9 | QED | SAS | Affinity |
|---|---|---|---|
| QED | 1.00 | -0.22 | 0.12 |
| SAS | -0.22 | 1.00 | 0.02 |
| Affinity | 0.12 | 0.02 | 1.00 |
| **ZINC15** | **QED** | **SAS** | **Affinity** |
| QED | 1.00 | -0.04 | -0.01 |
| SAS | -0.04 | 1.00 | 0.04 |
| Affinity | -0.01 | 0.04 | 1.00 |
| **PubChem** | **QED** | **SAS** | **Affinity** |
| QED | 1.00 | -0.18 | 0.19 |
| SAS | -0.18 | 1.00 | -0.06 |
| Affinity | 0.19 | -0.06 | 1.00 |

Notably, as the number of properties increases (e.g., more than 5), the assumption of independence may become less reasonable due to the potential rise in inter-property correlations. In such cases, more advanced modeling strategies that explicitly consider property interactions could be beneficial. However, it is common practice in multi-objective molecular design to focus on a small set of key properties.

## C.2    The Results of Surrogate Models

We assess the performance and reliability of the trained surrogate models using parity plots (Fig.C.1) and calibration curves (Fig.C.2). The parity plots demonstrate strong alignment between predicted and ground-truth values across all datasets and properties, with $R^2$ scores consistently above 0.8, indicating high regression accuracy. The accompanying histograms show that the predictions cover a wide and balanced range of property values, while the residual plots below each panel reveal no major systematic bias, with residuals centered around zero. These observations confirm that the models are not only accurate but also generalizable across molecular variations. The calibration curves further show that the predicted uncertainties are well calibrated, with AUCE values below 0.1 in most cases, suggesting that the uncertainty estimates meaningfully reflect the actual prediction errors. Together, these results confirm that the surrogate models provide accurate and trustworthy property predictions, supporting their use in uncertainty-aware reward construction.

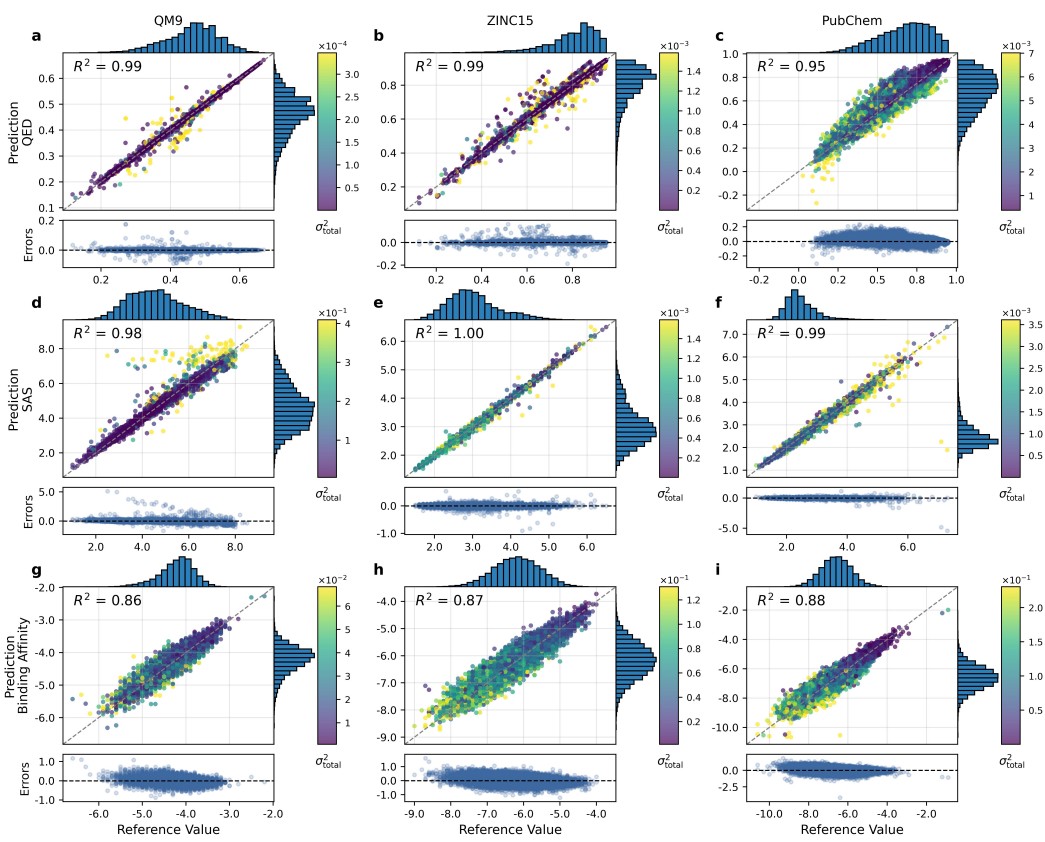

Figure C.1: **Parity plots comparing ground-truth property values with predictions from surrogate models.** The panels visualize results across three datasets (QM9, ZINC15, PubChem) and three properties (QED, SAS, binding affinity). Each point represents a molecule in the test set, colored by the predicted total uncertainty $\sigma^2_{\text{total}}$. The histograms indicate the distribution of predicted and true values. Scatter plots below each parity plot show residual errors. $R^2$ is reported in each plot to indicate prediction accuracy. $R^2$ values above 0.8 are generally considered to reflect strong predictive performance.

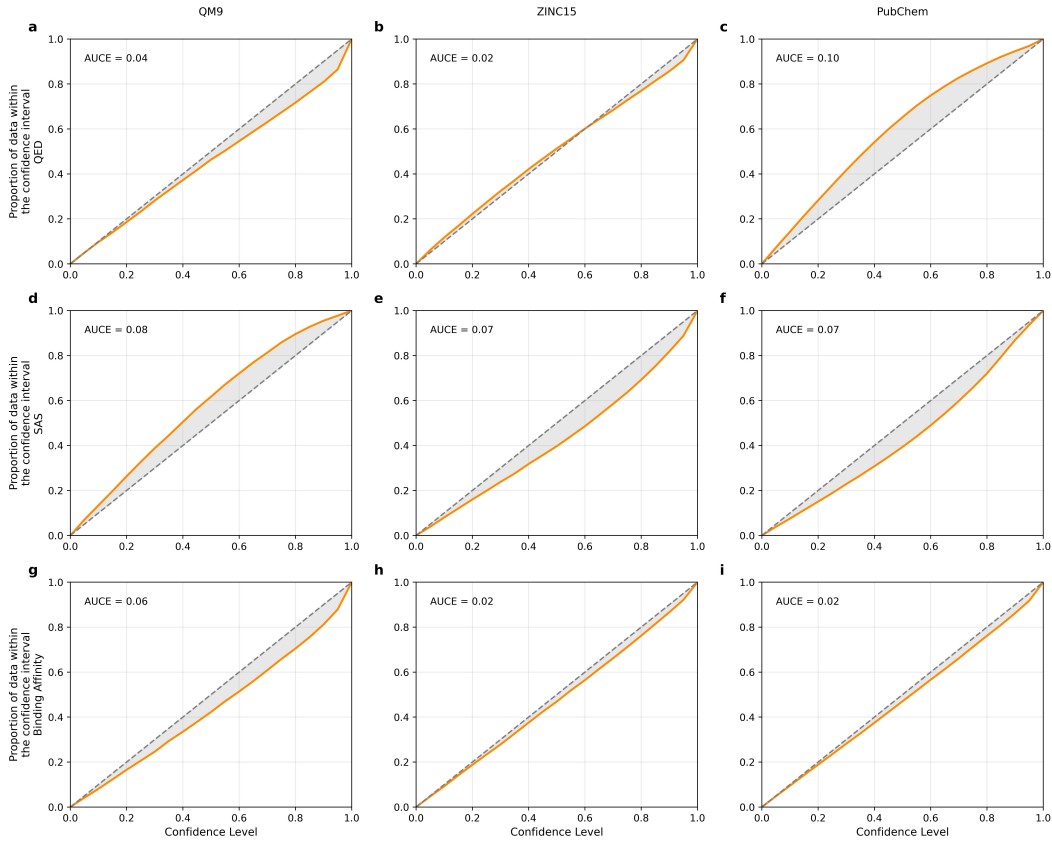

Figure C.2: **Confidence-based calibration curves (orange) of our surrogate models.** The plots show calibration performance on the test set across three datasets (QM9, ZINC15, PubChem) and three properties (QED, SAS, binding affinity). AUCE shown in each panel, quantifies miscalibration—lower AUCE values indicate better uncertainty calibration. The gray region denotes the deviation from ideal calibration (diagonal dashed line). AUCE values below 0.1 are generally considered indicative of well-calibrated uncertainty estimates.

## C.3 Extended Discussion on Baseline Comparisons

Similar to many multi-objective optimization problems, our binary reward design for these baselines creates a sparse and discontinuous optimization landscape. Molecules that are close to meeting the cutoffs but fail in one property receive no learning signal, making it difficult for the policy to gradually improve. This effect is exacerbated on challenging datasets where valid or near-optimal molecules are rare in the initial stages of training. As a result, during the training of some baselines, nearly all sampled molecules receive zero reward, leading to slow convergence and limited exploration.

SFT-PG approaches the problem from a different angle by minimizing the distributional gap between the model's generated property distributions and an ideal target distribution. While this avoids the brittleness of hard thresholds, it still suffers from indirect optimization. Without explicit instance-level reward feedback, the model may align distribution statistics while continuing to produce low-quality individual samples. This is because the training objective only penalizes aggregate discrepancies rather than enforcing quality at the individual molecule level. As a result, the model can satisfy distributional targets by generating a large number of average-quality samples, even if many of them fail to meet practical thresholds. This limitation is further exacerbated in settings with noisy or imperfect surrogate labels, where lack of instance-level supervision prevents the model from learning fine-grained improvements.

In contrast, our uncertainty-aware RL framework addresses these limitations through a probabilistic reward function that estimates the likelihood of each molecule satisfying individual property con-

straints. This yields several advantages. First, it provides denser and more informative gradient signals, allowing the policy to improve even for partially successful molecules. Second, it enhances sample efficiency and convergence speed by avoiding reward sparsity. Third, by incorporating both aleatoric and epistemic uncertainty, the model learns to prioritize high-confidence regions of chemical space, which is especially important under noisy or limited data.

Furthermore, we report additional baseline comparisons to further validate the effectiveness of our method. Table C.2 presents results obtained using a weighted sum of objective values for the baselines, rather than binary indicators, to enable more comprehensive comparisons. Table C.3 summarizes results from various optimization methods on the QM9 dataset, including Training-Free Guidance (TFG) [65], Best-of-N [66], Sequential Monte Carlo (SMC) [67], PILOT [68], and DiffSMol [69]. Additionally, Table C.4 reports experiments on two groups of real-valued properties on the QM9 dataset, further validating the proposed method. Overall, the results show that our method consistently achieves superior performance across all settings.

Table C.2: **Baseline results with a weighted-sum formulation of objectives on the QM9 dataset.**

| Method | Val (%) (↑) | Uni (%) (↑) | Nov (%) (↑) | VUN (%) (↑) | ASta (%) (↑) | MSta (%) (↑) | Top (%) (↑) |
|---|---|---|---|---|---|---|---|
| SFT-PG | $90.68 \pm 1.35$ | $96.12 \pm 0.24$ | $99.51 \pm 0.20$ | $86.73 \pm 1.22$ | $99.29 \pm 0.15$ | $95.92 \pm 0.43$ | $26.28 \pm 1.37$ |
| DDPO-SF | $91.02 \pm 1.68$ | $96.23 \pm 0.41$ | $99.39 \pm 0.11$ | $87.05 \pm 1.01$ | $99.35 \pm 0.22$ | $95.56 \pm 0.67$ | $26.32 \pm 1.18$ |
| DDPO-IS | $91.23 \pm 1.55$ | $95.27 \pm 0.87$ | $99.33 \pm 1.18$ | $86.33 \pm 0.79$ | $99.28 \pm 0.43$ | $89.18 \pm 0.37$ | $26.98 \pm 0.89$ |
| DPOK | $90.56 \pm 0.43$ | $\mathbf{96.91} \pm \mathbf{0.79}$ | $99.62 \pm 0.08$ | $87.43 \pm 0.67$ | $99.02 \pm 0.18$ | $95.48 \pm 0.64$ | $26.02 \pm 1.92$ |
| **Ours** | $\mathbf{98.17} \pm \mathbf{0.07}$ | $90.90 \pm 0.72$ | $\mathbf{99.63} \pm \mathbf{0.04}$ | $\mathbf{88.90} \pm \mathbf{0.68}$ | $\mathbf{99.87} \pm \mathbf{0.03}$ | $\mathbf{99.17} \pm \mathbf{0.27}$ | $\mathbf{28.33} \pm \mathbf{0.61}$ |

Note: Each model generates 2,000 molecules per run. Results are averaged over three independent runs and reported as mean ± 95% confidence interval. Higher values (bold) indicate better performance.

Table C.3: **More baseline comparisons on QM9 dataset.**

| Method | Val (%) (↑) | Uni (%) (↑) | Nov (%) (↑) | VUN (%) (↑) | ASta (%) (↑) | MSta (%) (↑) | Top (%) (↑) |
|---|---|---|---|---|---|---|---|
| TFG | $91.02 \pm 0.31$ | $\mathbf{97.10} \pm \mathbf{0.22}$ | $\mathbf{99.70} \pm \mathbf{0.10}$ | $88.12 \pm 0.37$ | $99.08 \pm 0.16$ | $95.67 \pm 0.60$ | $25.41 \pm 0.87$ |
| Best-of-N | $94.53 \pm 0.36$ | $93.02 \pm 0.17$ | $99.41 \pm 0.24$ | $87.41 \pm 0.38$ | $99.12 \pm 0.16$ | $96.56 \pm 0.52$ | $26.62 \pm 0.85$ |
| SMC | $95.11 \pm 0.23$ | $92.34 \pm 0.76$ | $99.69 \pm 0.52$ | $87.55 \pm 0.46$ | $99.47 \pm 0.12$ | $97.51 \pm 0.27$ | $26.95 \pm 0.83$ |
| PILOT | $95.20 \pm 0.23$ | $91.92 \pm 0.61$ | $99.61 \pm 0.38$ | $87.17 \pm 0.43$ | $99.13 \pm 0.27$ | $97.14 \pm 0.35$ | $27.01 \pm 0.72$ |
| DiffSMol | $94.12 \pm 0.28$ | $92.88 \pm 0.31$ | $99.59 \pm 0.16$ | $87.10 \pm 0.26$ | $99.01 \pm 0.29$ | $95.97 \pm 0.48$ | $26.51 \pm 0.74$ |
| **Ours** | $\mathbf{98.17} \pm \mathbf{0.07}$ | $90.90 \pm 0.72$ | $99.63 \pm 0.04$ | $\mathbf{88.90} \pm \mathbf{0.68}$ | $\mathbf{99.87} \pm \mathbf{0.03}$ | $\mathbf{99.17} \pm \mathbf{0.27}$ | $\mathbf{28.33} \pm \mathbf{0.61}$ |

Note: Each model generates 2,000 molecules per run. Results are averaged over three independent runs and reported as mean ± 95% confidence interval. Higher values (bold) indicate better performance.

Table C.4: **Results on two groups of real-valued properties on the QM9 dataset.**

| Group | Method | Val (%) (↑) | Uni (%) (↑) | Nov (%) (↑) | VUN (%) (↑) | ASta (%) (↑) | MSta (%) (↑) | Top (%) (↑) |
|---|---|---|---|---|---|---|---|---|
| Group 1 | W/O RL | $89.78 \pm 0.73$ | $\mathbf{97.98} \pm \mathbf{0.36}$ | $99.56 \pm 0.48$ | $87.58 \pm 0.76$ | $99.06 \pm 0.37$ | $86.02 \pm 0.32$ | $32.07 \pm 0.23$ |
| | **Ours** | $\mathbf{95.96} \pm \mathbf{1.27}$ | $94.80 \pm 0.73$ | $\mathbf{99.58} \pm \mathbf{0.52}$ | $\mathbf{90.59} \pm \mathbf{0.45}$ | $\mathbf{99.35} \pm \mathbf{0.07}$ | $\mathbf{92.14} \pm \mathbf{0.37}$ | $\mathbf{39.84} \pm \mathbf{0.33}$ |
| Group 2 | W/O RL | $87.89 \pm 0.43$ | $\mathbf{95.32} \pm \mathbf{0.41}$ | $99.39 \pm 0.62$ | $83.27 \pm 0.53$ | $99.01 \pm 0.29$ | $86.23 \pm 0.65$ | $29.41 \pm 0.92$ |
| | **Ours** | $\mathbf{94.47} \pm \mathbf{0.19}$ | $95.16 \pm 0.53$ | $\mathbf{99.44} \pm \mathbf{0.16}$ | $\mathbf{89.39} \pm \mathbf{0.57}$ | $\mathbf{99.52} \pm \mathbf{0.43}$ | $\mathbf{92.26} \pm \mathbf{0.49}$ | $\mathbf{36.74} \pm \mathbf{1.13}$ |

Note: Each model generates 2,000 molecules per run. Results are averaged over three independent runs and reported as mean ± 95% confidence interval. Higher values (bold) indicate better performance. Group 1: HOMO–LUMO gap, HOMO energy, and LUMO energy. Group 2: Dipole moment ($\mu$), Heat capacity ($C_v$), and Isotropic polarizability ($\alpha$).

## C.4 Extended Discussion on Ablation Studies

To assess the contribution of each component in our uncertainty-aware RL framework, we conduct ablation studies that isolate their individual effects. Results show that uncertainty-guided reward scaling is essential for stability and reliability. It prevents the model from overfitting to spurious high scores and ensures consistent improvement across multiple objectives. The diversity penalty further encourages structural exploration by discouraging repeated generation of similar molecules, leading to a broader search and a higher yield of top-performing candidates. Dynamic thresholding allows the model to adaptively respond to varying property distributions, avoiding premature convergence caused by rigid cutoffs. Lastly, the reward bonus effectively sharpens selection pressure toward high-quality molecules, enhancing validity while preserving uniqueness and novelty. Each component contributes to a different dimension of generation, and their combination yields robust and pharmaceutically meaningful outputs.

## C.5 Free Energy Perturbation–based Affinity Prediction

To complement the docking results with a more rigorous assessment of ligand–protein binding strength, free energy perturbation calculations were applied to estimate the binding free energy ($\Delta G_{\text{bind}}$) of the four selected compounds.

Using exponential averaging based on the Zwanzig equation, we computed binding free energy as the difference between the ligand's free energy in the bound (complex) and unbound (ligand-only). All simulations were run in OpenMM.

As shown in the Table C.5, the predicted binding free energy values ranged from –4.14 to –4.29 kcal·mol$^{-1}$, corresponding to low-micromolar binding affinities. Importantly, the three designed compounds exhibited binding affinities that are comparable to or modestly better than the known EGFR inhibitor. While not dramatically superior, these results lend additional support to the docking-based predictions and suggest that the newly generated candidates have acceptable and potentially favorable binding characteristics.

Table C.5: **Free energy perturbation–based affinity prediction.**

| Compound | $\Delta G_{\text{complex}}$ (kcal/mol) | $\Delta G_{\text{ligand}}$ (kcal/mol) | $\Delta G_{\text{bind}}$ (kcal/mol) | **Predicted $K_d$ ($\mu$M)** |
|---|---|---|---|---|
| Known inhibitor | -19.5 | -15.4 | -4.14 | ~1.2 $\mu$M |
| Molecule I | -20.3 | -16.0 | -4.29 | ~0.89 $\mu$M |
| Molecule II | -20.0 | -15.7 | -4.26 | ~1.0 $\mu$M |
| Molecule III | -20.2 | -15.9 | -4.28 | ~0.9 $\mu$M |

Molecule I: the generated molecule shown in Fig. 4b of the main text. Molecule II: the generated molecule shown in Fig. 4c of the main text. Molecule III: the generated molecule shown in Fig. 4d of the main text.

## C.6 Extended Discussion on MD and ADMET Prediction

While the main text highlights the general trend of MD and ADMET results for generated molecules, several specific observations warrant further discussion. In the MD simulations, all three top-ranked candidates formed stable complexes with EGFR over the full 10,000 ps production trajectories. One candidate (Fig. 4c in main manuscript) exhibited excellent structural stability, maintaining a low RMSD relative to the equilibrated apo conformation, indicative of strong and persistent binding. Another (Fig. 4d in main manuscript) showed minimal conformational fluctuations across the simulation and preserved critical hydrogen bonds within the ATP-binding site, reinforcing its potential as a potent inhibitor.

Beyond structural dynamics, in silico pharmacokinetic profiling of these candidates using ADMET-AI revealed consistently favorable drug-like properties. All three molecules exhibited high predicted human intestinal absorption, low blood-brain barrier permeability, and no violations of Lipinski's rule. Importantly, one of the molecule (Fig. 4d in main manuscript) showed low predicted hepatotoxicity and high metabolic stability, while another (Fig. 4b in main manuscript) scored favorably across P450 inhibition panels, suggesting low potential for metabolic drug-drug interactions.

These extended findings further support that top molecules generated by our framework are not only structurally valid and property-optimized, but also promising in terms of downstream pharmacological viability. Incorporating such analyses into post-generation filtering can substantially improve the relevance of AI-designed molecules for real-world drug discovery workflows.

## C.7 Visualization for Generated Molecules

We visualize representative candidate molecules generated by RL-guided diffusion models pre-trained on different datasets in Fig. C.3. For optimized model, we rank all generated molecules based on their multi-objective uncertainty scores. The top eight molecules with the highest scores from each model are shown. This selection strategy ensures that the displayed candidates are not only high-quality in terms of predicted properties, but also robust with respect to the model's uncertainty.

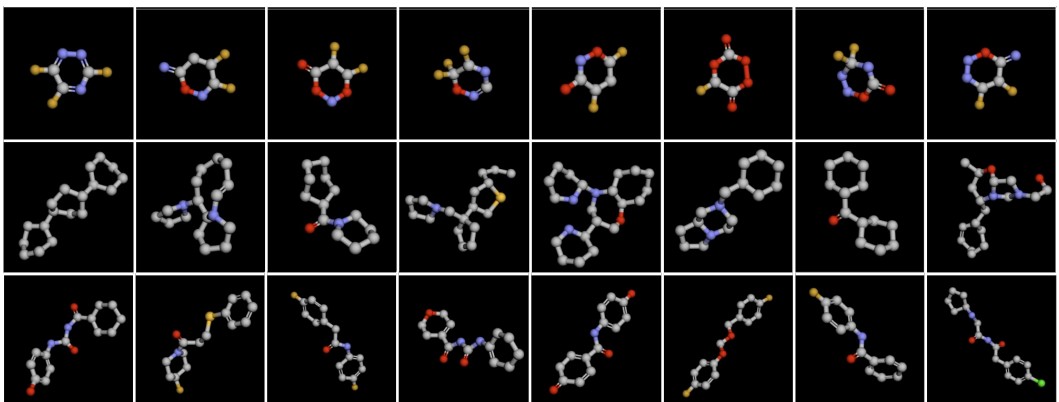

Figure C.3: **Representative candidate molecules generated by RL-guided diffusion models.** Each model was pre-trained on one of the three datasets (QM9, ZINC15, PubChem) and generated 2,000 molecules. The top 8 molecules selected by composite properties are shown: first row: QM9, second row: ZINC15, third row: PubChem.

## C.8 Analysis on Other Diffusion Models

Table C.6 summarizes the performance of our RL-guided optimization framework across other two diffusion models (GeoLDM and GFMDiff) on the QM9 dataset. Our method consistently improves the key evaluation metrics over the non-RL baselines, including validity, novelty and top molecules. This demonstrate that our method generalizes well across different diffusion models.

Table C.6: **Performance of other diffusion models and our method on QM9 datasets.**

| Model | Method | Val (%) (↑) | Uni (%) (↑) | Nov (%) (↑) | VUN (%) (↑) | ASta (%) (↑) | MSta (%) (↑) | Top (%) (↑) |
|---|---|---|---|---|---|---|---|---|
| GeoLDM | W/O RL | $91.98 \pm 0.17$ | $95.13 \pm 0.64$ | $99.71 \pm 0.07$ | $87.25 \pm 0.52$ | $99.08 \pm 0.16$ | $95.67 \pm 0.60$ | $85.40 \pm 0.00$ |
| | **Ours** | $95.11 \pm 1.24$ | $94.12 \pm 0.31$ | $99.73 \pm 0.13$ | $89.28 \pm 1.57$ | $99.12 \pm 0.17$ | $95.54 \pm 0.49$ | $87.08 \pm 0.02$ |
| GFMDiff | W/O RL | $94.33 \pm 0.17$ | $96.34 \pm 0.81$ | $96.11 \pm 1.03$ | $87.35 \pm 1.72$ | $99.75 \pm 0.03$ | $98.23 \pm 0.03$ | $74.50 \pm 0.00$ |
| | **Ours** | $96.72 \pm 1.27$ | $95.93 \pm 0.94$ | $96.15 \pm 0.21$ | $89.21 \pm 1.39$ | $99.77 \pm 0.10$ | $98.35 \pm 0.56$ | $75.01 \pm 0.01$ |

Note: Each model generates 2,000 molecules per run. Results are averaged over three independent runs and reported as mean ± 95% confidence interval. "W/O RL" denotes vanilla diffusion models without RL. "Val", "Uni", and "Nov" represent the percentages of valid, unique, and novel molecules, respectively. "VUN" is their joint metric computed as Val × Uni × Nov, representing the percentage of molecules that are simultaneously valid, unique, and novel. "ASta" and "MSta" denote atom-level and molecule-level stability. "Top" indicates the proportion of generated molecules that simultaneously satisfy all three property constraints, using relaxed cutoffs (QED > 0.4, SAS < 8, and binding affinity < –4.5) to avoid missing potentially useful candidates. All metrics are reported as percentages, and higher values indicate better performance.

