# OpenReview forum: "Uncertainty-Aware Multi-Objective Reinforcement Learning-Guided Diffusion Models for 3D De Novo Molecular Design"
_NeurIPS.cc/2025/Conference — NeurIPS 2025 poster_

### Official Review · Reviewer_2S15 · 2025-06-24

**Clarity:** 4
**Significance:** 3
**Originality:** 3
**Rating:** 5
**Confidence:** 4

**Summary:**

This paper presents a reinforcement learning framework integrating uncertainty for de novo molecular design diffusion models. The proposed method leverages predictive uncertainty from surrogate models to construct multi-objective reward functions, enabling the generation of drug-like molecules that balance key properties such as drug-likeness, synthetic accessibility, and binding affinity. Evaluations across benchmark datasets and multiple multi-property aggregation methods show performance gains over existing RL baselines. The approach is further validated through molecular dynamics simulations and ADMET profiling.

**Questions:**

- What are the implications of assuming independence in Eq. 3? How reasonable is this assumpton when a big number of properties are considered in multi-objective optimization?
- How much influence do the authors think the way they aggregate the multi-objective task to a single-objective in the baselines has on the results?
- Why do the authors think the pre-trained models on ZINC have so low permance? Why is still possible to rescue these models with Reinforcement Learning and reach quite good performance?

**Ethical Concerns:**

["NO or VERY MINOR ethics concerns only"]

**Final Justification:**

The authors have successfully addressed my comments. I raise the score accordingly.

**Limitations:**

-	The MD simulation time parameters are poorly reported as steps or frames. Reporting total physical simulation time would be much more informative. It's unclear right now, if the reopted simuations are equilibrated.
-	Improvements in already good docking score do not necessarily correlate with better affinity. Confirming high affinity with a more reliable affinity prediction method (such as Free Energy Perturbation) on the 4 molecules in Figure 4 would strengthen the paper.
-	Given that most of the work on fine-tuning diffusion models for optimized chemical spaces has been done in the context of diffusion guidance, I think it would strengthen the contribution to compare against some these works. Some examples are [1,2,3].

[1] PILOT: equivariant diffusion for pocket-conditioned de novo ligand generation with multi-objective guidance via importance sampling. Julian Cremer, Tuan Le, Frank Noé, Djork-Arné Clevert and Kristof T. Schütt. Chem. Sci., 2024,15, 14954-14967. DOI	https://doi.org/10.1039/D4SC03523B

[2] Unified Guidance for Geometry-Conditioned Molecular Generation. Sirine Ayadi, Leon Hetzel, Johanna Sommer, Fabian Theis, Stephan Günnemann. 38th Conference on Neural Information Processing Systems (NeurIPS 2024). https://arxiv.org/abs/2501.02526

[3] Generating 3D small binding molecules using shape-conditioned diffusion models with guidance. Ziqi Chen, Bo Peng, Tianhua Zhai, Daniel Adu-Ampratwum and Xia Ning. Nat Mach Intell 7, 758–770 (2025). https://doi.org/10.1038/s42256-025-01030-w

**Paper Formatting Concerns:**

.

**Quality:**

3

**Strengths And Weaknesses:**

Strengths:

-	Paper is clear and well written.
-	Simple method outperforming baselines.
-	Authors extensively compare both against other RL baselines and multi-property optimization methods.

Weaknesses:

-	Relevance and presentation of MD results (addressed during revisions).
-	Authors do not compare against stablished methods in the molecular domain (addressed during revisions)

---

> ### Author Rebuttal · Authors · 2025-07-31
>
> We sincerely thank the reviewer for the detailed and thoughtful feedback. We are pleased that the reviewer found our paper to be clear, well written, and of good technical quality. We especially appreciate the recognition that our method is effective in outperforming baselines, and that the experimental evaluation includes extensive comparisons with both RL baselines and multi-property optimization approaches. The positive assessment across quality, clarity, significance, and originality is encouraging and valuable to us.
>
> The reviewer’s concerns primarily relate to the presentation and interpretation of molecular dynamics results, the assumptions and design choices in our multi-objective reward formulation, the aggregation strategy used in baseline comparisons, and the comparison with existing diffusion-guided molecular optimization methods.
>
> We provide detailed responses below.
>
> **1. What are the implications of assuming independence in Eq. 3? How reasonable is this assumpton when a big number of properties are considered in multi-objective optimization?**
>
> **The independence assumption in Eq. 3 allows us to represent the multi-objective reward as a product of individual property-specific probabilities**, where each probability corresponds to the likelihood that a generated molecule exceeds a predefined cutoff for a given property. These probabilities are estimated using uncertainty-aware predictions from Chemprop, and their product provides a smooth and differentiable reward signal suitable for RL.
>
> In our study, we focus on three key properties: QED (drug-likeness), SAS (synthetic accessibility), and binding affinity. These properties are designed to reflect different and complementary aspects of molecule quality. **Empirically, we find that these properties are only weakly correlated**. As shown in the correlation matrix below, all pairwise Pearson correlation coefficients between the properties are less than 0.3 in magnitude, indicating low correlation and supporting the independence assumption in this context.
>
> **Correlation Matrix**
>
> |QM9     | QED   | SAS   | AFFINITY |
> |--------|-------|-------|----------|
> | **QED**     | 1.00  | -0.22 | 0.12     |
> | **SAS**     | -0.22 | 1.00  | 0.02     |
> | **AFFINITY** | 0.12  | 0.02  | 1.00     |
>
> |ZINC15    | QED   | SAS   | AFFINITY |
> |--------|-------|-------|----------|
> | **QED**     | 1.00  | -0.04 | -0.01    |
> | **SAS**     | -0.04 | 1.00  | 0.04     |
> | **AFFINITY** | -0.01 | 0.04  | 1.00     |
>
> |PubChem | QED   | SAS   | AFFINITY |
> |--------|-------|-------|----------|
> | **QED**     | 1.00  | -0.18 | 0.19     |
> | **SAS**     | -0.18 | 1.00  | -0.06    |
> | **AFFINITY** | 0.19  | -0.06 | 1.00     |
>
> We agree that as the number of properties increases (e.g., more than 5), the assumption of independence may become less reasonable due to the potential rise in inter-property correlations. In such cases, more advanced modeling strategies that explicitly consider property interactions could be beneficial. However, it is common practice in multi-objective molecular design to focus on a small set of key properties.
>
>
> **2.How much influence do the authors think the way they aggregate the multi-objective task to a single-objective in the baselines has on the results?**
>
> Aggregating multiple objectives into a single scalar reward, as done in baseline methods, may introduce challenges in RL. For example, it may lead to sparse early reward signals, resulting in slow learning, and can fail to provide positive feedback for “promising candidate molecules” that are close to satisfying all objectives but fall just short on one. This limits the model’s ability to explore near-optimal solutions and may reduce the effectiveness of optimization.
>
> However, in practical molecular design tasks, it is often required that generated molecules simultaneously meet all specified property thresholds, as only such molecules are typically retained for subsequent stages of evaluation. Therefore, using an aggregation strategy actually aligns with common practice in drug discovery workflows.
>
> In addition, to the best of our knowledge, there are very few baseline methods in the molecular generation literature that are directly compatible with our task setting. Among the limited available approaches, the most commonly used strategies for multi-objective optimization are weighted sum and gradient alignment. We included both in our ablation analysis, and our method consistently outperforms them.
>
>
> **3.Why do the authors think the pre-trained models on ZINC have so low permance? Why is still possible to rescue these models with Reinforcement Learning and reach quite good performance?**
>
> The relatively low performance of the pre-trained models on the ZINC dataset may stem from the fact that the general training objective used during pretraining is not sufficiently aligned with the highly specific multi-property criteria required for high-quality molecule generation. As a result, the pre-trained models may learn to capture overall data distribution patterns but fail to prioritize regions of chemical space corresponding to desirable property combinations.
>
> Through RL optimization, the model is further refined by using desirable property combinations collected from the test set as conditioning inputs, together with a goal-directed reward function to guide generation. This setup explicitly directs the model to generate molecules that satisfy the required constraints, enabling it to effectively adjust its generation strategy. As a result, the quality of generated molecules under specific desirable conditions is significantly improved, leading to better overall performance.
>
> **4.The MD simulation time parameters are poorly reported as steps or frames. Reporting total physical simulation time would be much more ......**
>
> To improve clarity, we have revised the reporting of MD simulation time throughout the paper. All relevant figures and descriptions now use total physical simulation time in picoseconds (ps), rather than steps or frames. This change provides a more interpretable and standard measure of simulation progress.
>
> To confirm equilibration, we re-ran the MD simulations for the four selected compounds and analyzed the RMSD of each protein–ligand complex. The RMSD profiles showed clear stabilization after an initial adjustment period, indicating that the systems reached equilibrium before further analysis. This confirms the reliability of the trajectories used for evaluating binding stability (as the RMSD values plateaued within a stable range of approximately 0.20–0.30 nm without signs of sustained drift or abrupt fluctuations).
>
>
> **5.Improvements in already good docking score do not necessarily correlate with better affinity. Confirming high affinity with a more reliable affinity prediction method ......**
>
> We appreciate the reviewer’s suggestion and agree that improvements in docking scores should be supported by more rigorous affinity predictions. To that end, we applied Free Energy Perturbation (FEP) calculations to estimate the binding free energy (ΔG bind) of the four compounds
>
> Using exponential averaging based on the Zwanzig equation, we computed binding free energy as the difference between the ligand’s free energy in the bound (complex) and unbound (ligand-only). All simulations were run in OpenMM.
>
> As shown in the table below, the predicted binding free energy values ranged from –4.14 to –4.29 kcal/mol, corresponding to low micromolar binding affinities. Importantly, the three designed compounds exhibited binding affinities that are comparable to or modestly better than the known inhibitor. While not dramatically superior, these results lend additional support to the docking-based predictions and suggest that the newly generated candidates have acceptable and potentially favorable binding characteristics.
>
> | Compound            | ΔGcomplex (kcal/mol) | ΔGligand (kcal/mol) | ΔGbind (kcal/mol) | Predicted Kd (μM) |
> |---------------------|----------------------|----------------------|-------------------|-------------------|
> | known inhibitor     | -19.5                | -15.4                | -4.14             | ~1.2 μM           |
> | mol from b (QM9)    | -20.3                | -16.0                | -4.29             | ~0.89 μM          |
> | mol from c (ZINC15) | -20.0                | -15.7                | -4.26             | ~1.0 μM           |
> | mol from d (PubChem)| -20.2                | -15.9                | -4.28             | ~0.9 μM           |
>
> **6.Given that most of the work on fine-tuning diffusion models for optimized chemical spaces has been done in the context of diffusion guidance, I think it would strengthen the contribution to compare against some these works. Some examples are [1,2,3].**
>
> We have carefully reviewed all three papers. We note that the second paper does not appear to provide publicly available source code, which makes reproducing their method for comparison difficult. As for the first and third papers, we observe that their approaches rely on large-scale pretraining on protein–ligand complex datasets, whereas our method is pretrained only on molecular data. As a result, it is challenging to directly apply their models to our dataset.
>
> Besides, their methods are designed for conditional generation based on protein inputs. We considered the possibility of using our target proteins as input to their pretrained models to perform a practical comparison. However, the distribution of our target proteins is substantially different from those used in their training data, making this approach infeasible in practice.
>
> That said, we appreciate the reviewer’s suggestion and will continue to explore whether these works can be meaningfully included in future comparisons, as well as investigate other potential baselines that could better align with our task setting.
>
> All the updated figure and analysis will be included in the camera-ready version of the paper.

---

> > ### Comment · Reviewer_2S15 · 2025-08-03
> >
> > I thank the authors for adressing most of my points. However I consider one last fundamental point has not been fully adressed.
> >
> > Regarding the last point (6), I don't fully follow why diffusion guidance is not applicable here. For example, why cannot Algorithm 1 in [1] be applicable in this setting? Isn't it just a importance reweighting step that be done during inference of your diffusion models? To be clear, the critical point I think has not been adressed here is how do diffusion guidance methods compare to your multi-objective RL?
> >
> > Finally, could the authors please update the manuscript pdf with the updated figures and results?

---

> > > ### Author Response · Authors · 2025-08-04
> > >
> > > We sincerely thank the reviewer for their thoughtful follow-up and for acknowledging our previous responses.
> > >
> > > You are absolutely right. The initial reply emphasized dataset construction challenges but did not sufficiently address the question from a methodological perspective. We fully agree with your observation and have since conducted additional experiments based on the publicly available implementations from the two diffusion guidance-based baseline methods. The results of this comparison are shown in the table below. As the table illustrates, our method outperforms the baselines in this setting.
> > >
> > > | Method   | Val (%) ↑     | Uni (%) ↑     | Nov (%) ↑     | VUN (%) ↑     | ASta (%) ↑    | MSta (%) ↑    | Top (%) ↑     |
> > > |----------|---------------|---------------|---------------|---------------|---------------|---------------|---------------|
> > > | PILOT[1]   | 95.20 ± 0.23  | 91.92 ± 0.61  | 99.61 ± 0.38  | 87.17 ± 0.43  | 99.13 ± 0.27  | 97.14 ± 0.35  | 27.01 ± 0.72  |
> > > | DiffSMol[3]| 94.12 ± 0.28  | 92.88 ± 0.31  | 99.59 ± 0.16  | 87.10 ± 0.26  | 99.01 ± 0.29  | 95.97 ± 0.48  | 26.51 ± 0.74  |
> > > | **Ours** | 98.17 ± 0.07  | 90.90 ± 0.72  | 99.63 ± 0.04  | 88.90 ± 0.68  | 99.87 ± 0.03  | 99.17 ± 0.27  | 28.33 ± 0.61  |
> > >
> > > Additionally, we would be very willing to update our figures and results in the manuscript. We have carefully reviewed the **NeurIPS 2025 FAQ for Authors**, and understand that we are not permitted to upload a revision of our paper or supplementary materials during the rebuttal/discussion phase. However, we are fully willing to incorporate the updated figures and results in the **camera-ready version**. If any clarification regarding the figures or results would be helpful at this stage, we would be glad to provide additional explanation.
> > >
> > > Below, we quote the relevant statement from the **NeurIPS 2025 FAQ for Authors**:
> > >
> > > **Can we upload a revision of our paper during the rebuttal/discussion period?** No revisions are allowed until the camera-ready stage.
> > >
> > > **Can we upload a revision of the supplementary materials during the rebuttal/discussion period?** No. You may revise it for the camera-ready stage.
> > >
> > > Thank you very much!
> > >
> > > [1] PILOT: equivariant diffusion for pocket-conditioned de novo ligand generation with multi-objective guidance via importance sampling. Julian Cremer, Tuan Le, Frank Noé, Djork-Arné Clevert and Kristof T. Schütt. Chem. Sci., 2024,15, 14954-14967.
> > >
> > > [3] Generating 3D small binding molecules using shape-conditioned diffusion models with guidance. Ziqi Chen, Bo Peng, Tianhua Zhai, Daniel Adu-Ampratwum and Xia Ning. Nat Mach Intell 7, 758–770 (2025).

---

> > > > ### Comment · Reviewer_2S15 · 2025-08-06
> > > >
> > > > Thank you for the extra results and clarifications. My concerns have been resolved and I increase my score accordingly.

---

> > > > > ### Author Response · Authors · 2025-08-06
> > > > >
> > > > > Dear Reviewer 2S15,
> > > > > Many thanks for your valuable feedback. We’re glad to hear that your concerns have been resolved and that you are willing to increase your score accordingly. Much appreciated!
> > > > >
> > > > > Thanks.

---

### Official Review · Reviewer_qJWH · 2025-07-03

**Clarity:** 2
**Significance:** 1
**Originality:** 1
**Rating:** 3
**Confidence:** 4

**Summary:**

The paper proposes an uncertainty-aware RL framework to post-train a molecular diffusion model towards multiple objectives. A surrogate model predicts uncertainty estimation, which is used to reweight reward functions to keep objectives balanced. Experiments on three molecule generation benchmarks as well as MD simulations and ADMET profiling tasks validates the effectiveness of the framework.

**Questions:**

Please refer to the weaknesses.

**Ethical Concerns:**

["NO or VERY MINOR ethics concerns only"]

**Final Justification:**

Based on the rebuttal responses and discussions, I raised my score from 2 to 3. While the authors resolved part of my concerns, I'm still concerning about the novelty and the mixed performance among metrics.

**Limitations:**

Yes.

**Quality:**

2

**Strengths And Weaknesses:**

Strengths: the framework introduces predictive uncertainty into multi-objective RL framework.



Weakness:

1. Although there are many terms and components in the title, the technical details of most components are not novel, making the title and the paper confusing.

   * Diffusion Models for 3D De Novo Molecular Design: simply a conditional EDM, which has been deployed by hundreds of papers.
   * Uncertainty-Aware: the predictive uncertainty is obtained by an integration with threshold of predicted mean and variance via D-MPNN, which is simple, black-box and not strict. There are more established methods in uncertainty quantification, such as [1].
   * Multi-Objective: the aggregation assumes independency and simply multiplies the single-objective rewards. Again, there are tons of papers in multi-objective optimization.
   * Reinforcement Learning: seems like the major contribution of the paper. The policy update method resembles PPO, so the only novel part is the reward design. What is the real difference between proposed method and baselines?

2. The comparison between proposed method and baselines is unclear. There are many methods to handle multi-objective RL, but the authors ignore them and the baselines are treated in a single-objective manner with a ridiculous binary reward, which is clearly unfair. Also, please include inference-time reward-guided sampling methods as alternative baselines, such as Sequential Monte Carlo (SMC) and Best-of-N.

3. For experiments, QM9 is a dataset with internal 19 types of molecule properties, including some chemical and thermodynamic quantities. I strongly encourage to include this task (predicting multiple real-valued molecule properties) and compare with conditional molecule generation baselines mentioned in GeoLDM [2].



[1] Huang, K., Jin, Y., Candes, E., & Leskovec, J. (2023). Uncertainty quantification over graph with conformalized graph neural networks. *Advances in Neural Information Processing Systems*, *36*, 26699-26721.

[2] Xu, M., Powers, A. S., Dror, R. O., Ermon, S., & Leskovec, J. (2023, July). Geometric latent diffusion models for 3d molecule generation. In *International Conference on Machine Learning* (pp. 38592-38610). PMLR.

---

> ### Author Rebuttal · Authors · 2025-07-31
>
> We appreciate the reviewer’s recognition of our contribution, including the integration of uncertainty into multi-objective RL and comprehensive validation across multiple benchmarks, MD simulations, and ADMET analysis.
>
> The concerns raised by the reviewer mainly stem from: the novelty, the baseline setup, and the experimental design on the QM9 dataset. Below, we respond to these concerns in detail.
>
> **1. Although there are many ...**
>
> While some components have been explored separately, our main contribution is their integration into a unified, end-to-end framework for 3D therapeutic molecule design. Besides, we address some common challenges in using RL to optimize diffusion models in our task.
>
> **Diffusion Models for 3D De Novo ...**
>
> We chose to build our framework on EDM because **it is a seminal work in applying diffusion models to 3D molecular generation**. Based on our prior extensive literature review, we found that only around 10 papers have extended EDM, most of which were published at top-tier venues such as NeurIPS and ICLR. This demonstrates that EDM is a well-recognized and foundational framework, making it a suitable and credible starting point for our work.
>
> Furthermore, most existing works based on EDM focus on improving generation quality or conditioning only on the target protein. In contrast, **our study is distinct in its goal of designing therapeutic molecules that satisfy multiple drug-relevant properties simultaneously**. This is a **practically motivated** setting that aims to produce candidate compounds suitable for synthesis and experimental testing by our collaborating chemists.
>
> We also acknowledge the reviewer’s mention of alternative diffusion backbones, such as GeoLDM. To demonstrate the generality of our method, **we additionally evaluated our framework on other advanced diffusion models, including GeoLDM and GFMDiff**, as detailed in **Appendix C.6**.
>
>
> **Uncertainty-Aware: the predictive ...**
>
> We appreciate the reviewer’s suggestion regarding ConFormalized (CF)-GNN. While it may offer theoretical advantages, CF-GNN has practical limitations for RL, including slow inference due to calibration set access and unstable reward signals caused by threshold effects. Given these factors, **CF-GNN-based methods are not well suited for RL deployment**.
>
> While we acknowledge the reviewer’s concerns regarding the use of D-MPNN, we would like to emphasize that **we conducted a rigorous evaluation of our surrogate models to ensure their reliability**. As shown in **Appendix C.1**, we provided detailed assessments including parity plots, calibration curves, and quantitative metrics such as R² and AUCE. All surrogate models achieved high R² scores (> 0.8), indicating strong predictive accuracy, and low Area Under the Calibration Error (AUCE) values (< 0.05), demonstrating well-calibrated uncertainty estimates. **These results confirm that our D-MPNN-based models are both accurate and reliable**.
>
> In this context, the D-MPNN-based approach compensates well for any theoretical limitations through strong empirical performance, making it a more practical and stable choice for RL deployment.
>
>
> **Multi-Objective: the aggregation ...**
>
> We acknowledge the reviewer’s concern regarding the independence assumption in our reward aggregation. To support this design choice, we computed the Pearson correlation coefficients between all pairs of properties and found that all coefficients are below 0.3. **This indicates weak correlations among the properties and justifies the assumption of independence in our experimental setup**.
>
> By aggregating the multi-objective reward as the product of individual property-specific rewards, we simplify the reward formulation while preserving a smooth and interpretable optimization landscape. **This design aligns with the goal of enabling efficient and stable policy learning in the RL framework**. Empirically, this approach has proven to be highly effective, as demonstrated by our results.
>
> We agree that multi-objective optimization is a well-studied area. To ensure a fair and comprehensive comparison, **we included 16 commonly used multi-objective optimization methods** in our ablation study. Our method consistently outperformed these other optimization strategies, further validating the effectiveness of our approach.
>
>
> **Reinforcement Learning: seems ...**
>
> We thank the reviewer for recognizing the contributions of our work, particularly in RL and reward design. To the best of our knowledge, at the time of submission, **our work is the first to present an end-to-end framework that integrates RL, diffusion models, 3D molecular generation, and uncertainty-based multi-objective optimization**. This design has not been explored in prior molecular generation literature. As mentioned earlier, our framework addresses a real-world application in designing novel EGFR inhibitors as alternatives to known inhibitors, which aligns with the NeurIPS application theme.
>
> In terms of technical innovation, beyond incorporating uncertainty into the reward function, we addresses several challenges in using RL to diffusion models in our tasks, including mode collapse, premature convergence, and sparse or unstable reward signals. To this end, we introduce three auxiliary mechanisms:
>
> **a. Reward Boosting Mechanism**: Designed to mitigate sparse reward signals in the early stages of training, this mechanism incorporates soft auxiliary rewards based on molecular validity, uniqueness, and novelty. It enables the policy to quickly identify viable regions in chemical space, accelerates learning, and improves sample efficiency.
>
> **b. Diversity Penalty Mechanism**: Aims to prevent mode collapse and premature convergence by penalizing structural redundancy within a batch of generated molecules. This mechanism encourages exploration of diverse chemical structures, helping the model avoid overfitting to narrow high-reward regions and improving coverage of the chemical space.
>
> **c. Dynamic Cutoff Strategy**: Addresses reward saturation or instability caused by static thresholds during training. By adaptively updating the property-specific cutoffs based on the evolving distribution of generated molecules, this mechanism ensures that the reward signal remains discriminative and informative throughout training. As a result, it supports stable learning dynamics and better alignment with multi-objective constraints.
>
> The contribution of each module is validated in our ablation study (see **Table 2** in the main paper).
>
>
> **2. The comparison between proposed ...**
>
> Our task focuses on RL-guided optimization during the training phase of diffusion models. In this context, and following established practices in the field, we adopt gradient-based policy optimization methods such as PPO. These methods are generally preferred for direct integration with neural networks in an end-to-end fashion, as they allow for efficient backpropagation, unlike many other RL methods. Among gradient-based RL methods, the most common strategies for handling multiple objectives include weighted sum and Pareto-based methods such as PCGrad. As mentioned earlier, we have already included these strategies in our ablation analysis.
>
> We acknowledge that the use of binary rewards in baseline designs may introduce challenges in RL training, such as sparse early rewards and lack of feedback for near-miss candidates. However, this design aligns with the practical goal of therapeutic molecule design, where only molecules that meet all required property thresholds are considered viable. In real-world drug discovery pipelines, it is common to retain only compounds that satisfy all desired criteria. Therefore, **while a binary reward may not be ideal from an RL optimization perspective, it is a principled and task-aligned formulation rather than an unfair or incorrect one**.
>
> We also agree that it is difficult to identify prior works that are directly comparable in this our setting. Nonetheless, we appreciate the reviewer’s suggestion and have Best-of-N and SMC as baselines. The table below show that our method outperforms both approaches, supporting its effectiveness.
>
> | Method     | Val (%) ↑     | Uni (%) ↑     | Nov (%) ↑     | VUN (%) ↑     | ASta (%) ↑    | MSta (%) ↑    | Top (%) ↑     |
> |------------|---------------|---------------|---------------|---------------|---------------|---------------|---------------|
> | Best-of-N  | 94.53 ± 0.36  | 93.02 ± 0.17  | 99.41 ± 0.24  | 87.41 ± 0.38  | 99.12 ± 0.16  | 96.56 ± 0.52  | 26.62 ± 0.85  |
> | SMC        | 95.11 ± 0.23  | 92.34 ± 0.76  | 99.69 ± 0.52  | 87.55 ± 0.46  | 99.47 ± 0.12  | 97.51 ± 0.27  | 26.95 ± 0.83  |
> | **Ours** | 98.17 ± 0.07  | 90.90 ± 0.72  | 99.63 ± 0.04  | 88.90 ± 0.68  | 99.87 ± 0.03  | 99.17 ± 0.27  | 28.33 ± 0.61  |
>
>
> **3. For experiments, QM9 is a dataset ...**
>
> Under the application theme, our work is focused on designing novel molecules that can serve as alternatives to known EGFR inhibitors. In this context, we prioritized the properties of binding affinity, synthetic accessibility, and drug-likeness. **In contrast, many of the typical chemical and thermodynamic properties included in QM9, such as HOMO/LUMO energy levels and heat capacity at constant volume, are either unrelated or only indirectly related to drug discovery and are not considered key decision-making factors in therapeutic molecule design.** Therefore, these properties were not prioritized in our study.
>
> We used QM9 as part of our experiments primarily because its simplicity and small molecular size (up to 9 heavy atoms) make it a practical starting point for model development and early-stage validation. Our task formulation does not depend on the specific molecular properties provided by the QM9 dataset.
>
> That said, incorporating property prediction tasks into molecular design could be an interesting direction. We appreciate the reviewer’s suggestion and will consider exploring this in future work.

---

> > ### Comment · Reviewer_qJWH · 2025-08-04
> >
> > I thank the authors for the response.
> >
> > 1. I still hold the opinion that simply integrating these existing components seems to be a limited contribution. As is also acknowledged by the authors, this paper does not propose any novel component individually besides the RL reward design. In particular, the paper leverages simple diffusion baselines, naive uncertainty estimation and multi-objective aggregation. The arguments that these simple UQ methods work does not seem convincing to me, and the proposed method does not consistently outperform all other multi-objective RL baselines.
> >
> > 2. I appreciate the comparison with SMC and best-of-N. Unfortunately, the authors did not address my concern that the comparison between the proposed method and other baselines is inconsistent - simple binary reward is used for other baselines while the precise numerical reward is used for the proposed method. In contrast, a common practice in previous papers is to use weighted sum of the numerical rewards of each objective.
> >
> > 3. The purpose of proposing using other molecular properties is to further verify that the proposed method actually work, while the authors did not conduct the experiment. Moreover, QM9 is a toy dataset with very small molecules. The paper fails to provide any result on larger datasets with larger molecules, e.g., MOSES or GEOM-DRUG, thus cannot validate the universality and scalability of the proposed method.
> >
> > Given the above concerns, I will keep my rating.

---

> > > ### Author Response · Authors · 2025-08-07
> > > **New experiment results to address your concerns**
> > >
> > > **1.1 Novelty of the framework and the advantages of the UQ approach**
> > >
> > > As acknowledged by the reviewer, our **reward design is novel.** We highlight three further aspects of novelty in our work.
> > >
> > > First, the standalone evaluation of our UQ module has sufficiently demonstrated its reliability, as shown in **Appendix C.1**. Moreover, its integration within our framework leads to significant performance improvements, as shown in Table 2 of the manuscript. Our UQ module, though simple in form, addresses a largely underexplored area in molecular generation. We believe its simplicity offers both practicality and interpretability.
> > >
> > > Second, we propose several mechanisms aimed at addressing persistent issues in RL–based optimization of diffusion models, such as mode collapse, premature convergence, and stagnation, which are challenges that remain in prior works.
> > >
> > > Third, we understand the reviewer’s concern that integrating existing modules may not, in itself, constitute a strong contribution. However, we respectfully argue that in application-driven domains like molecular generation, system-level integration of AI modules is a meaningful form of innovation. It requires careful engineering to align multiple components toward a complex real-world goal. Our framework, to the best of our knowledge, is the first of its kind in molecular generation and directly addresses key practical challenges by integrating these existing and new modules developed in this study.
> > >
> > > **1.2 Performance**
> > >
> > > In the comparison with other multi-objective tasks (Table 2 in the manuscript), our method achieved the best performance across all key evaluation metrics, including Val, VUN, ASta, MSta, and Top. We understand the Uni and Nov are not the highest, but they are calculated relative to Val, and lower values do not necessarily indicate weaker generative capability. The metrics that best reflect generative performance are Val and VUN, on which our method achieved the best results.
> > >
> > > **2. Incorporating binary rewards allows for diverse baseline comparisons.**
> > >
> > > **We adopted the reviewer’s suggestion and used the weighted sum of the numerical values of each objective for the baselines.** The results are shown below (CI omitted for space). To clarify, the baselines in Table 1 of the manuscript used binary rewards, and the newly added version with precise values is shown below. For all analyses of the 16 methods in Table 2 of the manuscript, precise numerical rewards were used throughout.
> > >
> > > | Method   | Val(%)↑ | Uni(%)↑ | Nov(%)↑ | VUN(%)↑ | ASta(%)↑ | MSta(%)↑ | Top(%)↑ |
> > > |----------|---------|---------|---------|----------|-----------|-----------|----------|
> > > | SFT-PG   | 90.68   | 96.12   | 99.51   | 86.73    | 99.29     | 95.92     | 26.28    |
> > > | DDPO-SF  | 91.02   | 96.23   | 99.39   | 87.05    | 99.35     | 95.56     | 26.32    |
> > > | DDPO-IS  | 91.23   | 95.27   | 99.33   | 86.33    | 99.28     | 89.18     | 26.98    |
> > > | DPOK     | 90.56   | 96.91   | 99.62   | 87.43    | 99.02     | 95.48     | 26.02    |
> > > | **Ours** | 98.17   | 90.90   | 99.63   | 88.90    | 99.87     | 99.17     | 28.33    |
> > >
> > > **3. Consideration of other properties and datasets**
> > >
> > > To further verify the proposed method, we also conduct experiments on two groups of real-valued properties on the QM9 dataset, as the reviewer suggested. Group 1: HOMO-LUMO Gap, HOMO energy, and LUMO energy. Group 2: Dipole Momen, Heat Capacity, Isotropic Polarizability. The results are shown below.
> > >
> > > | Group   | Method | Val(%)↑ | Uni(%)↑ | Nov(%)↑ | VUN(%)↑ | ASta(%)↑ | MSta(%)↑ | Top(%)↑ |
> > > |---------|--------|---------|---------|---------|----------|-----------|-----------|----------|
> > > | Group1  | W/O RL | 89.78   | 97.98   | 99.56   | 87.58    | 99.06     | 86.02     | 32.07    |
> > > |         | Ours   | 95.96   | 94.80   | 99.58   | 90.59    | 99.35     | 92.14     | 39.84    |
> > > | Group2  | W/O RL | 87.89   | 95.32   | 99.39   | 83.27    | 99.01     | 86.23     | 29.41    |
> > > |         | Ours   | 94.47   | 95.16   | 99.44   | 89.39    | 99.52     | 92.26     | 36.74    |
> > >
> > > Our framework has been evaluated on 3 representative datasets of different scales, as reported in Table 1 of the main paper. In addition to QM9, we conducted experiments on the ZINC15 and PubChem datasets.
> > >
> > > For the **ZINC15 dataset**, it contains over **190k molecules**, with up to **26 heavy atoms per molecule**. The **PubChem dataset** includes over **440k molecules**, with up to **50 heavy atoms per molecule**. These details are also reported in Appendix B.2.2.
> > >
> > > However, for the two datasets the reviewer mentioned, the MOSES is primarily based on SMILES representations and is typically used for 1D molecular tasks. It does not provide 3D molecular structures, whereas our task focuses on 3D molecular design. As for the GEOM, its processed form contains about 200k molecules, and the majority of these molecules contain only between 20 and 30 heavy atoms, which does not exceed the molecular complexity present in the PubChem dataset we used.

---

> > > > ### Author Response · Authors · 2025-08-08
> > > > **Follow-Up on Rebuttal**
> > > >
> > > > Dear Reviewer qJWH,
> > > > Thank you for your comments. We submitted our rebuttal to your comments yesterday and would like to kindly invite you to take a look. If you have any remaining concerns, please let us know before the discussion period ends today so we can address them. We hope our responses, highlighting the novelty of our work, extensive experiments, and clarifications, help you reassess our paper.
> > > > Thanks.
> > > > Authors

---

> > > > > ### Comment · Reviewer_qJWH · 2025-08-08
> > > > >
> > > > > Thank you for your additional comments. I'm updating my score reflecting the author's efforts.

---

> > > > > > ### Author Response · Authors · 2025-08-08
> > > > > > **Thank you**
> > > > > >
> > > > > > Dear Reviewer qJWH, Thank you for your insightful feedback. We are pleased that we were able to address your comments and sincerely appreciate your willingness to consider raising the score.
> > > > > >
> > > > > > The authors

---

### Official Review · Reviewer_cfze · 2025-07-10

**Clarity:** 3
**Significance:** 3
**Originality:** 3
**Rating:** 5
**Confidence:** 3

**Summary:**

In this work RL is incorporated in the training of guided-diffusion models to facilitate generation of 3D molecules satisfying
multiple properties, waiving the need to have differentiable surrogate models of the properties of interest. A conditional diffusion model is first trained on the molecular dataset then further optimized using RL with an uncertainty-based multi-objective reward to encourage the generation of molecules satisfying multiple constraints.

**Questions:**

1) Clarify task directions in line 133. Does it refer to minimization or maximization of a property?

2) In the forward process there is conditioning on C in Figure 1 caption, is this needed? This does not match the description of forward process in Appendix A.1.

3) In RL-guided training, it is explained that property combinations satisfying predefined cutoffs (QED > 0.4, SAS < 8, binding affinity < -4.5) are used as conditions.
In generation, how the combinations of property values (QED, SAS, Binding affinity) are chosen for conditioning?

4) Does RL-guidance help only with better conditional generation or it is helping with property optimization. i.e., maximizing/minimizing properties? If the later is not the case, how can your method be altered for this purpose?

5) Based on Figure 3 (WO/RL vs W/RL), for ZINC15 dataset, my guess is that the failure of other methods, in satisfying all three properties (TOP % Metric), lies in failing the QED property which is also reflected in the validity metric (Table 1). How is ZINC15 dataset different from other datasets that could justify this significant difference in performance?

6) (Out of curiosity) How does the RL-guided optimization after pretraining on each dataset, affects the quality of conditional generation for samples used for pretraining?

**Ethical Concerns:**

["NO or VERY MINOR ethics concerns only"]

**Final Justification:**

All my questions were addressed by the authors.

**Quality:**

3

**Strengths And Weaknesses:**

The paper has nice narrative and experiments to demonstrate the capabilities of the proposed model.

---

> ### Author Rebuttal · Authors · 2025-07-31
>
> We sincerely thank the reviewer for the thoughtful and constructive feedback. We are glad that the reviewer found our paper technically sound, with a clear presentation and well-designed experiments. We also appreciate the recognition of the method’s quality, originality, and significance, as well as the acknowledgment that our experimental results effectively demonstrate the model’s capabilities.
>
> The reviewer’s concerns mainly center around clarifications related to task directions, the conditioning scheme in the forward process, the selection of property combinations during generation, and the role of RL in conditional generation versus property optimization. We thank the reviewer for raising these important points, and we address each concern in detail below.
>
>
> **1. Clarify task directions in line 133. Does it refer to minimization or maximization of a property?**
>
> **Yes**. Our experiments consider three key properties: Quantitative Estimate of Drug-likeness (QED), Synthetic Accessibility Score (SAS), and binding affinity. In our task, the objective is for all three properties of a molecule to simultaneously exceed predefined cutoff values, rather than optimizing a single property in isolation. Once a molecule satisfies all three cutoffs, we prefer QED to be as high as possible (indicating better drug-likeness), SAS to be as low as possible (indicating easier synthesis in laboratory conditions), and binding affinity to be as low as possible (indicating stronger binding to the target protein).
>
> Accordingly, in line 133, the task direction for QED is set to "maximize", while the directions for SAS and binding affinity are set to "minimize".
>
> **2. In the forward process there is conditioning on C in Figure 1 caption, is this needed? This does not match the description of forward process in Appendix A.1.**
>
> Thank you for pointing this out. This is a typo in the figure. **The forward process in Figure 1 does not require conditioning**. We will correct this in the camera-ready version.
>
> **3. In RL-guided training, it is explained that property combinations satisfying predefined cutoffs (QED > 0.4, SAS < 8, binding affinity < -4.5) are used as conditions. In generation, how the combinations of property values (QED, SAS, Binding affinity) are chosen for conditioning?**
>
> In generation, we need to input the combination of multiple properties into the model to guide molecule generation. Since we consider multiple properties simultaneously, there are infinitely many possible combinations. To avoid feeding unreasonable or invalid property combinations into the model, **we first filter the test set using predefined cutoff values to obtain reasonable property combinations. We then sample from these filtered combinations to serve as conditional inputs for generation**.
>
> **4. Does RL-guidance help only with better conditional generation or it is helping with property optimization. i.e., maximizing/minimizing properties? If the later is not the case, how can your method be altered for this purpose?**
>
> Thank you for this insightful question. Our RL-guided training primarily improves conditional generation by encouraging the model to generate molecules that satisfy multiple property constraints simultaneously. We do not explicitly formulate the task as direct maximization or minimization of individual properties.
>
> This design choice is motivated by the fact that the properties (QED, SAS, binding affinity) we consider are often conflicting. For example, aggressively minimizing binding affinity (which is desirable from a binding perspective) may lead to complex or unrealistic molecules with poor synthetic accessibility or low drug-likeness. Optimizing a single property in isolation can therefore harm the overall quality and viability of the generated molecules.
>
> To avoid such trade-offs, our reward is defined as the probability that a generated molecule satisfies all predefined property cutoffs. These probabilities are estimated using uncertainty-aware predictions from surrogate models, resulting in a scalar reward value in the [0, 1] range. This approach enables us to softly integrate multi-objective constraints without explicitly requiring the maximization/minimization of any one property at the expense of others.
>
> That said, if one wishes to adapt our method for pure property optimization (e.g., maximizing binding affinity), the reward function can be modified accordingly. For instance, one could define the reward as a weighted combination of raw property scores or use property-specific gradients where feasible. However, this would require careful consideration of property conflicts and trade-offs.
>
> **5. Based on Figure 3 (WO/RL vs W/RL), for ZINC15 dataset, my guess is that the failure of other methods, in satisfying all three properties (TOP % Metric), lies in failing the QED property which is also reflected in the validity metric (Table 1). How is ZINC15 dataset different from other datasets that could justify this significant difference in performance?**
>
> The ZINC15 dataset differs from other datasets in two key aspects:
>
> 1)**ZINC15 is a drug-like molecular dataset**, in which molecules generally possess favorable pharmacological properties. As a result, it provides training samples that are inherently closer to the target distribution of the optimization task, making it easier for the model to generate molecules that meet the desired property constraints.
>
> 2)**Molecules in ZINC15 typically contain a moderate number of heavy atoms** and tend to exhibit less complex topological structures and conformational variability. This makes it easier for the model to preserve structural validity during both the noise perturbation and reverse generation processes.
>
> In contrast, QM9 is not a drug-oriented dataset, and many of its molecules lack desirable pharmacological characteristics. This increases the distributional shift and structural adaptation challenges the model must overcome during learning and optimization. As for PubChem, although its molecules often exhibit good drug-like properties, they frequently contain a large number of heavy atoms, complex functional groups, and intricate ring systems. These features result in a combinatorially large structural space, making the reverse diffusion process more prone to accumulating minor prediction errors, which can ultimately lead to structural collapse.
>
> **6. (Out of curiosity) How does the RL-guided optimization after pretraining on each dataset, affects the quality of conditional generation for samples used for pretraining?**
>
> Thank you for this thoughtful question. After pretraining, we further optimize the model using Reinforcement Learning (RL), where the conditional inputs are high-quality property combinations sampled from the test set. During both RL optimization and evaluation, the same set of high-quality combinations is used as conditioning input. Before RL, the model’s performance under these conditions was relatively limited. **With RL optimization, the model significantly improves its conditional generation quality**. For example, we observe higher validity and a greater proportion of top molecules. This indicates that RL effectively enhances the model’s ability to generate molecules under conditions related to the pretraining distribution. It also highlights a gap between exposure during pretraining and actual controllability under target conditions, which RL helps to close by directly optimizing generation performance under those conditions.
>
> Thank you again for your question and for your recognition of our work. Please feel free to share any further questions or suggestions you may have.

---

> > ### Author Response · Authors · 2025-08-06
> >
> > Dear Reviewer cfze,
> > Thank you very much for taking the time to read our manuscript. We would like to confirm whether our responses have fully addressed your comments and concerns. If you have any remaining questions or issues, we would be happy to provide further clarification or revisions.
> >
> > Thank you again for your time and consideration.

---

> > ### Comment · Reviewer_cfze · 2025-08-08
> > **Response to the rebuttal**
> >
> > Thanks for answering all my questions. I will raise my score to 5.

---

> > > ### Author Response · Authors · 2025-08-08
> > > **Thanks**
> > >
> > > Dear Reviewer cfze, Thank you for your valuable feedback. We’re delighted that your concerns have been addressed and truly appreciate your willingness to raise your score to 5.

---

### Official Review · Reviewer_Phx1 · 2025-07-21

**Clarity:** 3
**Significance:** 2
**Originality:** 3
**Rating:** 4
**Confidence:** 4

**Summary:**

The paper presents a methodologically sound pipeline, which systematically integrates three key stages: reward design, data sampling, and optimization using Proximal Policy Optimization (PPO). The authors conduct their analysis within a rigorous experimental setting, ensuring a fair and direct comparison against established baselines.

**Questions:**

1. How is uncertainty computed?

2. Is this method applicable to other architectures, such as GeoLDM?

3. How does the RL-based method compare to training-free guidance [1]?

[1] https://arxiv.org/abs/2209.15408

**Ethical Concerns:**

["NO or VERY MINOR ethics concerns only"]

**Final Justification:**

Since the method also performs well on GeoLDM, I am willing to raise my score.

**Limitations:**

Please check weakness.

**Quality:**

3

**Strengths And Weaknesses:**

Strengths
The paper presents a logically sound and well-structured methodological pipeline.

The experimental evaluation is rigorous, employing fair comparisons against relevant baselines.

A key contribution is the novel integration of an uncertainty metric directly into the reward function, which is a compelling idea.

Weaknesses
The primary weakness is the limited methodological novelty. The proposed framework is largely an assembly of well-established components, which may not meet the innovation threshold for this venue.

The choice to demonstrate the method on an explicit-density model (EDM) instead of a more challenging and widely-used latent diffusion model (like GeoLDM) limits the perceived impact. Applying guidance to latent models is a more difficult and generalizable problem, and the current choice may be seen as targeting a simpler use case. Demonstrating success on a latent model would significantly strengthen the contribution.

---

> ### Author Rebuttal · Authors · 2025-07-31
>
> We thank the reviewer for their thoughtful and constructive feedback. We are encouraged that the reviewer finds our pipeline to be logically sound and well-structured, and appreciates the rigorous experimental evaluation and fair comparisons against relevant baselines. We are particularly pleased that the integration of an uncertainty metric into the reward function is seen as a compelling and valuable contribution.
>
> The reviewer’s main concerns center on: 1. the novelty; 2. the choice of model architecture (EDM vs. GeoLDM); 3. the uncertainty computation; and 4. the comparison to training-free guidance methods.
>
> Below, we address each of these concerns in detail and provide clarifications and evidence.
>
> **1. The novelty**
>
> 1)To the best of our knowledge, at the time of submission, **our work is the first to propose an end-to-end framework that integrates Reinforcement Learning (RL), diffusion models, 3D molecular generation, and uncertainty-based multi-objective optimization.** This combination has not previously been explored in the molecular generation literature. In other application domains, RL has been used to optimize diffusion models, primarily for single-objective tasks such as improving aesthetic scores in image generation. Representative examples include DPOK: RL for Fine-tuning Text-to-Image Diffusion Models and DDPO: Training Diffusion Models with RL, which apply RL to guide diffusion-based image synthesis. Although these works are also not the first to combine RL with diffusion models, they have demonstrated strong performance and clear practical value in the context of image generation and were accepted at NeurIPS and ICLR main tracks.
>
> 2)In terms of technical innovation, beyond the novel integration of uncertainty quantification into the reward function to address multi-objective optimization, **we also tackle several challenges faced by RL in the context of optimizing diffusion models in our task**. These challenges include mode collapse, premature convergence, and stagnation during the learning process. To address these issues, we design three auxiliary mechanisms in our reward function:
>
> a.**Reward boosting mechanism**. This mechanism integrates additional constraints such as molecular validity, uniqueness, and novelty into the training objective in the form of soft rewards. This enhances the flexibility of the reward structure and explicitly encourages the model to generate chemically feasible molecules while optimizing their desired properties. Besides, it strengthens the reward signal during the early stages of training, allowing the policy to quickly identify viable regions of the chemical space and avoid getting stuck in the sparse-reward phase. Further implementation details were provided in **Appendix A.3**.
>
> b.**Diversity penalty mechanism**. The diversity penalty encourages structural exploration by penalizing overly similar molecules within a batch. This drives the policy to explore new regions of the chemical space and prevents premature convergence to narrow, high-reward clusters. From an RL perspective, this mechanism functions similarly to entropy regularization by promoting broader policy support and mitigating mode collapse.
>
> c. **Dynamic Cutoff Strategy**. When converting uncertain property predictions into probabilistic improvement signals, it is necessary to define a cutoff threshold for each property. In existing studies, these thresholds are typically set as static values. However, such static thresholds can quickly become misaligned with the evolving distribution of generated molecules during training, leading to reward saturation or sparsity. These issues may undermine training stability or result in premature convergence. To ensure that the reward signal remains both discriminative and informative throughout the training process, we propose a dynamic cutoff mechanism. This method adaptively computes property-specific cutoff values based on the shifting distribution of predicted values in each training segment. Specifically, we determine the quantile used for cutoff calculation using cosine interpolation between 50% and 10%, depending on the current stage of training. This quantile is then applied to the property distribution of newly generated molecules to compute updated cutoffs (top x% values). These adaptive cutoffs are subsequently used when evaluating the probabilistic improvement of generated molecules.
>
> The contribution of each module can be referenced from the ablation study presented in **Table 2** of the manuscript. The results show that our RL-optimized model consistently improves performance across multiple metrics and datasets. Notably, on the ZINC15 dataset, we observe substantial gains in key evaluation criteria: the validity of generated molecules increases by 70%, and the proportion satisfying all property constraints rises by 25%.
>
> We thank the reviewer for pointing this out and will incorporate the corresponding updates in the camera-ready version.
>
> 3)**Our work closely follows the Guidelines for Writing a Good NeurIPS Paper in relation to the application theme**:
>
> a.The guidelines state: “Application papers should describe your work on a ‘real’ as opposed to ‘hypothetical’ application; specifically, it should describe work that has direct relevance to, and addresses the full complexity of, solving a non-trivial problem.” Our work directly targets a clinically relevant and complex challenge in drug design. Specifically, we propose an AI-based framework for generating novel therapeutic molecules that simultaneously optimize multiple critical molecular properties, including drug-likeness, synthetic accessibility, and binding affinity to a target protein. As a result, we successfully discovered novel candidate molecules with performance comparable to known EGFR inhibitors. This work is part of a larger interdisciplinary collaboration. Currently, our partner chemists are synthesizing and experimentally validating the generated molecules to assess their real-world therapeutic potential.
>
> b. The guidelines also emphasize that “Authors are encouraged to convey insight about the problem, algorithms, and/or application,” and that “Techniques shown to be uniquely fitted to specific popular applications, leading to improved performance or more accurate solutions” are particularly valued. As discussed in our points 1) and 2), our method was carefully designed to match the specific challenges of 3D molecular generation and multi-objective drug discovery, satisfying these criteria.
>
> c. Finally, the guidelines state that “A NeurIPS application paper should be comparable in quality to papers in the corresponding application domain conference.” We believe our work meets this bar. Beyond the algorithmic contributions, we also developed a user-friendly interface to broaden accessibility. Users without programming experience can upload a target receptor structure and easily obtain candidate therapeutic molecules, as detailed in **Appendix D**.
>
>
> **2. The choice of model architecture (EDM vs. GeoLDM)**
>
> We chose to demonstrate our framework on the EDM because it is a seminal work in applying diffusion models to 3D molecular generation. Subsequent research efforts, such as GeoLDM, have been built upon this framework. Therefore, we consider EDM to be a robust and highly representative baseline, which makes it a natural and appropriate starting point for our experimental design.
>
> To further demonstrate the broad applicability of our framework, **we have also evaluated it on other more advanced diffusion models, including GeoLDM**. The corresponding results were reported in **Appendix C.6**, and for convenience, we include the relevant excerpt below. The results indicate our framework can be effectively applied across different 3D molecular diffusion models.
>
> **Table 1**
>
> | Model   | Method   | Val (%) ↑     | Uni (%) ↑     | Nov (%) ↑     | VUN (%) ↑     | ASta (%) ↑    | MSta (%) ↑    | Top (%) ↑     |
> |---------|----------|---------------|---------------|---------------|---------------|---------------|---------------|---------------|
> | GeoLDM  | W/O RL   | 91.98 ± 0.17  | 95.13 ± 0.64  | 99.71 ± 0.07  | 87.25 ± 0.52  | 99.08 ± 0.16  | 95.67 ± 0.60  | 85.40 ± 0.00  |
> |         | **Ours** | 95.11 ± 1.24  | 94.12 ± 0.31  | 99.73 ± 0.13  | 89.28 ± 1.57  | 99.12 ± 0.17  | 95.54 ± 0.49  | 87.08 ± 0.02  |
> | GFMDiff | W/O RL   | 94.33 ± 0.17  | 96.34 ± 0.81  | 96.11 ± 1.03  | 87.35 ± 1.72  | 99.75 ± 0.03  | 98.23 ± 0.03  | 74.50 ± 0.00  |
> |         | **Ours** | 96.72 ± 1.27  | 95.93 ± 0.94  | 96.15 ± 0.21  | 89.21 ± 1.39  | 99.77 ± 0.10  | 98.35 ± 0.56  | 75.01 ± 0.01  |
>
>
> **3. The uncertainty computation**
>
> The uncertainty is computed by combining aleatoric and epistemic components to estimate the total prediction variance. Based on this, the probability that a molecule’s property exceeds a target threshold is estimated using a Gaussian assumption. In multi-objective settings, the overall uncertainty is defined as the product of individual threshold probabilities across properties, assuming independence. This yields a unified reward signal reflecting the likelihood of satisfying all objectives simultaneously. Details were provided in **Appendix A.2**.
>
> **4. The comparison to training-free guidance methods**
>
> We established a Training-Free Guidance (TFG) baseline. The results are summarized in the table below and demonstrate that our method achieves better performance.
>
> **Table 2**
>
> | Method   | Val (%) ↑     | Uni (%) ↑     | Nov (%) ↑     | VUN (%) ↑     | ASta (%) ↑    | MSta (%) ↑    | Top (%) ↑     |
> |----------|---------------|---------------|---------------|---------------|---------------|---------------|---------------|
> | TFG      | 91.02 ± 0.31  | 97.10 ± 0.22  | 99.70 ± 0.10  | 88.12 ± 0.37  | 99.08 ± 0.16  | 95.67 ± 0.60  | 25.41 ± 0.87  |
> | **Ours** | 98.17 ± 0.07  | 90.90 ± 0.72  | 99.63 ± 0.04  | 88.90 ± 0.68  | 99.87 ± 0.03  | 99.17 ± 0.27  | 28.33 ± 0.61  |

---

> > ### Comment · Reviewer_Phx1 · 2025-08-03
> >
> > I appreciate the authors’ efforts to address my concerns. Since the method also performs well on GeoLDM, I am willing to raise my score. One question remains: why did the reinforcement-learning algorithm improve the validation metric? In my experience, the reward or loss doesn't contain validation, so validation performance typically declines after RL fine-tuning or guidance. Could you clarify this?

---

> ### Author Response · Authors · 2025-08-04
>
> Thank you very much for your feedback. We really appreciate that you are willing to raise the score.
>
> Regarding the clarification of the questions you raised, you are absolutely right that validation performance typically declines after reinforcement learning guidance. In our experiments, the primary reason that the reinforcement learning algorithm improves the validation metric is that we implicitly incorporate the validity metric into the reward function, namely the **reward boosting mechanism**. Specifically, as shown in equation (5) in the manuscript (which is also included below for convenience), we multiply \( U_{\text{multi}} \) by an additional factor \( R_{\text{bonus}} \). If the generated sample is valid, then \( U_{\text{multi}} \) will be scaled by the factor, which is greater than 1, to boost the reward. Otherwise, no such boosting is applied. The factor that results in the best performance is found via grid search.
>
> $$
> R_{\text{total}}(m; \delta_1, \ldots, \delta_k, t_{\text{episode}}) = U_{\text{multi}}(m; \delta_1, \ldots, \delta_k) \cdot R_{\text{bonus}}(m) - \lambda(t_{\text{episode}}) \cdot D(m) \tag{5}
> $$
>
> Under this mechanism, the generated valid molecules will consistently receive higher rewards, while invalid molecules will not receive any reward increase. In reinforcement learning, behaviors that lead to higher rewards are typically reinforced [1,2]. Therefore, as training progresses, the model is encouraged to explore regions of chemical space where molecules are more likely to be chemically valid, increasing the likelihood of generating valid structures. The related details can also be found in **Appendix A.3**.
>
> Another reason is that we also incorporate a **diversity penalty** in the reward function, specifically the D(m) term in Equation (5). Through this diversity penalty, the model is encouraged to explore broader areas of chemical space, leading to more diverse outputs. Such exploration increases the chance of discovering valid and property-optimized molecules, further contributing to the observed performance gain.
>
> Please feel free to let us know if you have any further questions, and we would be happy to provide clarification.
>
> If appropriate, we would be grateful if you could consider updating your score to reflect your current assessment.
>
> Thank you very much!
>
>
> [1] Sutton, R. S., & Barto, A. G. (2018). Reinforcement Learning: An Introduction. MIT Press.
>
> [2] Schulman, J., Wolski, F., Dhariwal, P., Radford, A., & Klimov, O. (2017). Proximal Policy Optimization Algorithms. arXiv preprint.

---

> > ### Author Response · Authors · 2025-08-06
> >
> > Dear Reviewer Phx1,
> > Thank you very much for your insightful feedback. We sincerely appreciate your willingness to consider raising the score.
> > We would like to confirm whether our latest responses have fully addressed your most recent comments. If you have any remaining concerns, we would be happy to further clarify or revise as needed.
> >
> > Thanks.

---

### Note · Authors · 2025-08-15

We thank the reviewers and the Area Chair for the constructive feedback and productive discussion. Following the exchange, all reviewers confirmed that our clarifications and new results addressed all their concerns and questions, and all four increased their scores.

**Reviewers highlighted the following strengths:**

1.The integration of Uncertainty Quantification (UQ) into the Reinforcement Learning (RL) framework for multi-objective optimization of 3D molecular diffusion models constitutes a valuable contribution. (Reviewer Phx1, qJWH)

2.Our methodological design is robust and well-structured, with clear writing, a coherent narrative, and solid technical quality. (Reviewer Phx1, cfze, 2S15)

3.Our approach delivers strong empirical performance, achieving superior results across datasets of varying scales and architectures, including both foundational and advanced diffusion models, and outperforming all baselines. (Reviewer Phx1, 2S15)

4.Our experimental evaluation is rigorous and fair, with comprehensive comparisons to RL baselines and a wide range of optimization methods. (Reviewer Phx1, 2S15)

5.The inclusion of downstream molecular dynamics simulations and ADMET profiling provides meaningful validation, further supporting the effectiveness and practical applicability of the proposed framework. (Reviewer qJWH, 2S15)

**To further substantiate these points, we provided the following additional experiments and clarifications during the rebuttal phase:**

1.Clarified the novelty: we proposed the first end-to-end framework that systematically integrates uncertainty-aware multi-objective RL with 3D molecular diffusion models for drug discovery, and introduced three mechanisms to address long-standing issues in RL-based diffusion model optimization.

2.Supported the effectiveness of our UQ method: we added pairwise correlation analyses among properties and clarified the reliability of the surrogate models for uncertainty prediction.

3.Expanded baseline comparisons: we included additional baselines, such as training-free guidance, inference-time reward-guided sampling, and diffusion-guidance methods.

4.Validated the generality: we offered results on two sets of real-valued properties, and clarified the strong performance across diffusion backbones and datasets.

In summary, our work addresses a complex real-world therapeutic design problem and aligns with the NeurIPS Application Track expectations for practical impact, methodological rigor and clarity.

---

### Decision · Program_Chairs · 2025-09-17

**Decision:**

Accept (poster)

**Comment:**

This paper proposes a 3D de-novo molecular design pipeline combining multi-objective RL with guided diffusion models.

All the reviewers are positive about the work, and the only reviewer who was negative, qJWH, also increased their score after the rebuttal. While there were several questions about the experiments, the authors gave convincing answers to all the questions and also added new supporting results.

One of the major criticisms against the paper was the lack of novelty. I do not think algorithmic novelty is necessary for the applications track. The paper does a great job in putting together a complex pipeline for 3d de-novo molecular design and that is worthy enough to be published!